# Asymmetric total synthesis of polycyclic xanthenes and discovery of a WalK activator active against MRSA

Min-Jing Cheng [1,2,6], Yan-Yi Wu [1,2,6], Hao Zeng [3,6], Tian-Hong Zhang [1,2], Yan-Xia Hu [1,2], Shi-Yi Liu [4], Rui-Qin Cui [4], Chun-Xia Hu [4], Quan-Ming Zou [3] ✉, Chuang-Chuang Li [5] ✉, Wen-Cai Ye [1,2] ✉, Wei Huang [4] ✉ & Lei Wang [1,2] ✉

The development of new antibiotics continues to pose challenges, particularly considering the growing threat of multidrug-resistant *Staphylococcus aureus*. Structurally diverse natural products provide a promising source of antibiotics. Herein, we outline a concise approach for the collective asymmetric total synthesis of polycyclic xanthene myrtucommulone D and five related congeners. The strategy involves rapid assembly of the challenging benzo-pyrano[2,3-a]xanthene core, highly diastereoselective establishment of three contiguous stereocenters through a *retro*-hemiketalization/double Michael cascade reaction, and a Mitsunobu-mediated chiral resolution approach with high optical purity and broad substrate scope. Quantum mechanical calculations provide insight into stereoselective construction mechanism of the three contiguous stereocenters. Additionally, this work leads to the discovery of an antibacterial agent against both drug-sensitive and drug-resistant *S. aureus*. This compound operates through a unique mechanism that promotes bacterial autolysis by activating the two-component sensory histidine kinase WalK. Our research holds potential for future antibacterial drug development.

The emergence of multidrug-resistant bacteria, specifically methicillin-resistant *Staphylococcus aureus* (MRSA), is recognized as a serious threat to public health[1,2]. For example, there were 323,700 cases and 106,00 deaths attributed to MRSA infections in the United States in 2017, resulting in nearly $1.7 billion in associated healthcare costs[3]. However, antibiotic resistance continues to rise, including resistance to vancomycin, daptomycin, and linezolid[4–6]. Therefore, the development of new antibacterial agents is of utmost urgency. A major focus of current antibiotic development is screening based on natural products[7–9]. Xanthenes are an important class of heterocyclic compounds in drug discovery and organic synthetic chemistry because of their broad range of biological activities, particularly their antibacterial activity[10–13], and diverse complex scaffolds. In recent years, xanthenes have attracted significant interest from the synthetic chemistry community, including groups such as Jauch[14], Ready[15], Siegel[16], Porco Jr[17], Martin[18], Pronin[19], Suzuki[20], and Gao[21]. To date, hundreds of xanthenes have been identified in nature[22–25]. As a paradigm, (+)-myrtucommulone D (**1**) (Fig. 1a) is an unusual angular benzopyrano[2, 3-a]xanthene

[1]State Key Laboratory of Bioactive Molecules and Druggability Assessment, Jinan University, Guangzhou 510632, P. R. China. [2]Center for Bioactive Natural Molecules and Innovative Drugs, and Guangdong Province Key Laboratory of Pharmacodynamic Constituents of TCM and New Drugs Research, College of Pharmacy, Jinan University, Guangzhou 510632, P. R. China. [3]National Engineering Research Center of Immunological Products, Department of Microbiology and Biochemical Pharmacy, College of Pharmacy, Army Medical University, Chongqing 400038, P. R. China. [4]Department of Medical Laboratory, Shenzhen People's Hospital (The Second Clinical Medical College, Jinan University; The First Affiliated Hospital, Southern University of Science and Technology), Shenzhen 518020 Guangdong, P. R. China. [5]Department of Chemistry, Shenzhen Grubbs Institute, Southern University of Science and Technology, Shenzhen 518055, P. R. China. [6]These authors contributed equally: Min-Jing Cheng, Yan-Yi Wu, Hao Zeng. ✉e-mail: qmzou2007@163.com; ccli@sustech.edu.cn; chywc@aliyun.com; whuang_sz@163.com; cpuwanglei@126.com

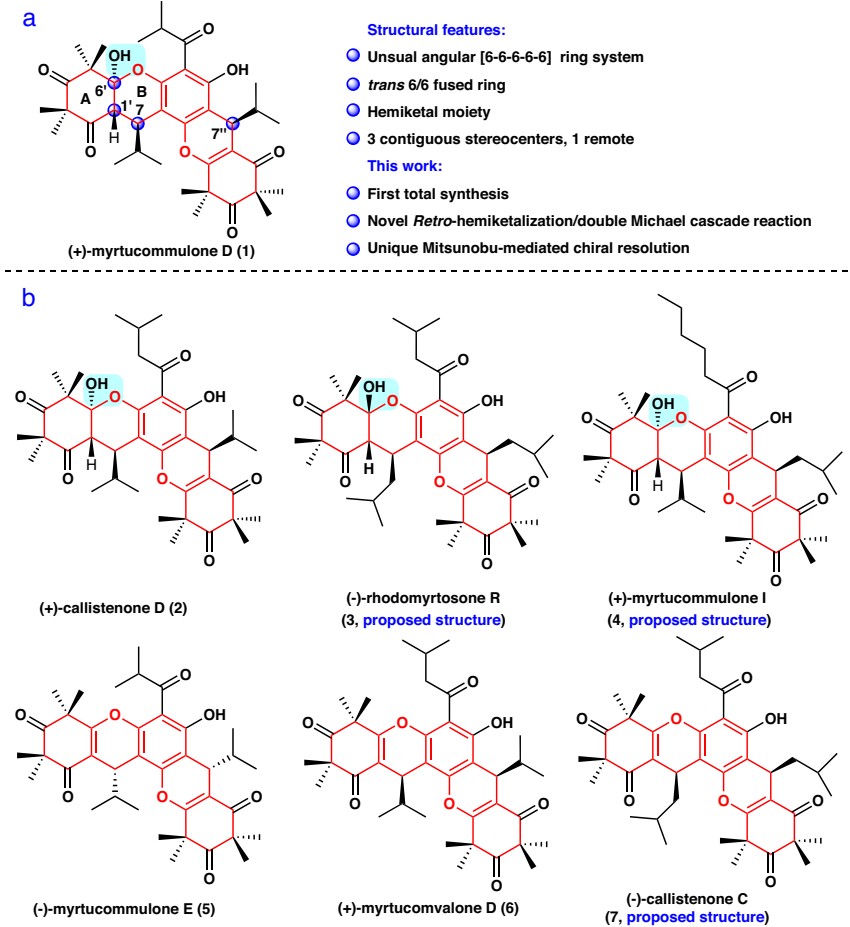

**Fig. 1 | Selected natural products with a common [6-6-6-6-6] ring system (highlighted in red), along with the structural characteristics and synthetic highlights. a** The structure characteristics of **1** and highlights of this synthetic work. **b** The structures of **2**–**7**. Blue box with a background refers to hemiketal moiety. The stereocenters are highlighted with a blue ball.

that was isolated in 2006 from *Myrtus communis* by Shaheen and colleagues[26], and showed significant activity against *S. aureus* and MRSA with a minimum inhibitory concentration (MIC) of 2 μg/mL[27]. Structurally, **1** is the first example of a xanthene derivative with a compact angular [6-6-6-6-6] pentacyclic skeleton (highlighted in red), containing a *trans* 6/6 fused ring and a C6′ hemiketal moiety. Additionally, it has four stereocenters, including three contiguous stereocenters (C7 → C1′ → C6′) and one quaternary stereocenter. Based on its structural complexity, the synthesis of **1** presents a significant challenge.

Notably, structurally relevant natural products have been also disclosed[26–34], such as (+)-callistenone D (**2**), (-)-rhodomyrtosone R (**3**), (+)-myrtucommulone I (**4**), (-)-myrtucommulone E (**5**), (+)-myrtucomvalone D (**6**), and (-)-callistenone C (**7**) (Fig. 1b), leading to an expansion of this group of promising natural products. Among these compounds, (±)-callistenone D (**2**) has been reported to display greater antibacterial activity against MRSA (MIC 1 μg/mL) compared with (+)- and (-)-myrtucomvalone D (**6**), and (±)-myrtucommulone E (**5**), which have weak to no activity (MIC > 128 μg/mL)[27]. As a result, it was hypothesized that the hemiketal moiety in **1**–**4** was critical for their antibacterial activity[27,33]. However, a systematic evaluation of their biological activity has been hindered by the relative scarcity of these compounds from natural sources[29]. Thus, there is a need for the development of a general strategy for the total synthesis of these compounds to accelerate the systematic study of their biological activities. To date, there are only a few studies concerning the asymmetric synthesis of potential synthetic precursors for these polycyclic

xanthenes, such as myrtucommulone B (**8**), rhodomyrtone B (**9**) and rhodomyrtone (**11**) (Fig. 2a). However, these studies either used excess amounts (3 equiv.) of the chiral Al−Li−BINOL (1,1′-bi-2-naphthol) complex, resulting in 62% enantiomeric excess (*ee*)[35]; or employed an expensive chiral phosphoric acid, which provided good enantioselectivity (90% *ee*) but required a long reaction time (6 days)[36]; or faced racemization due to *retro*-Friedel-Crafts reactions[37] (Fig. 2a). Furthermore, there have been few methods reported for the construction of such an angular [6-6-6-6-6] pentacyclic skeleton containing a hemiketal moiety[38]. Although several elegant synthetic strategies for constructing three contiguous stereocenters in xanthene have been developed[39–43], their configurations are different from that of **1**. Additionally, the absolute configurations of some compounds, such as **3**–**4** and **7**, have yet to be established, and the total synthesis of **1** and its related congeners have not been reported. In our continuing efforts toward the synthesis of bioactive polycyclic natural products[36,44–47], we report the inaugural collective asymmetric total syntheses of **1**–**3** and **5**–**7**, accomplished through a Knoevenagel/hemiketalization annulation, an intramolecular *retro*-hemiketalization/double Michael cascade reaction, and a unique Mitsunobu-mediated chiral resolution approach with a broad substrate scope. Through these syntheses, the absolute configurations of rhodomyrtosone R (**3**) and callistenone A (**7**) were determined[29,31]. More importantly, compound **22** was found to exhibit significant antibacterial activity against both drug-sensitive and drug-resistant *S. aureus* in vitro and in vivo. Further genetic and biochemical studies revealed that this compound promotes bacterial autolysis by activating the sensor histidine kinase WalK, making it a

**Fig. 2 | Reported attempts to asymmetric synthesis of polycyclic xanthene precursors and retrosynthetic analysis of polycyclic xanthenes. a** Reported attempts to asymmetric syntheses of **11** and its analogs. **b** Retrosynthetic strategy for myrtucommulone D and five related congeners.

promising antibacterial compound with an exceptional mechanism of action.

## Results and discussion

### Retrosynthetic strategy for 1–3 and 5–7

We envisioned the synthesis of polycyclic **1–3** and **5–7** by employing the highly convergent strategy depicted in Fig. 2b. Retrosynthetically, we anticipated that **5–7** could be first traced back to hemiketals **1–3**, respectively. The construction of the three contiguous stereocenters in the A and B rings of **1–3** would be a key issue of the synthesis. Disconnections between the C7 and $R_1$ group (highlighted in blue), the C6' and O at C1, and C1' and H were considered to address the stereochemistry of C7 in the B ring and achieve desired configurations at C6' and C1', respectively. The double Michael addition annulation cascade reaction has proven to be a powerful tool in organic synthesis[48]. Thus, we envisaged using the double Michael addition reaction for the construction of consecutive three stereocenters. The strategy suggested that the α,β-unsaturated enone **II** is a key intermediate in the synthesis. Subsequently, disassembly of **II** provided red fragment **12** and tetramethylxanthenone **III**, which would be coupled via a Knoevenagel reaction. However, the presence of the carbonyl groups in **III** would undergo a competing Knoevenagel reaction, which would make the anticipated reaction challenging. **III** would be installed through a Rieche reaction of **IV**. **IV** would be expected to be accessible from *rac*-**IV** with the inexpensive and available **13** (~40 \$/50 g) by a unique Mitsunobu-mediated chiral resolution. However, *rac*-**IV** would undergo the competitive chemo- and regioselective Mitsunobu reaction because of the presence of free phenol and the aromatic hydrogen in the precursor[49]. Moreover, the *ee* value may be affected because of

the involvement of a stereocenter in this reaction[50]. These issues would make the anticipated Mitsunobu-mediated resolution approach challenging.

### Asymmetric syntheses of (+)- and (-)-1 in addition to (+)- and (-)-5

We began our investigations with the synthesis of (+)- and (-)-myrtucommulone D (**1**), and (+)- and (-)-myrtucommulone E (**5**). As initial efforts to achieve the catalyzed asymmetric synthesis of the important intermediate isomyrtucommulone B (**14**) according to a previously reported protocol were not successful (see Supplementary Fig. 1.for details), we set out to develop a more robust route. To the best of our knowledge, chiral resolution via the diastereoisomers, which can be separated by silica gel column chromatography or recrystallization, has been widely used in asymmetric synthesis and pharmaceutical applications[51,52]. Thus, our synthesis of the key intermediate (+)- and (-)-**14** commenced with a chiral resolution between *rac*-**14** (prepared in two steps, see Supplementary Pages 18–20 for details) and the chiral reagents. After extensive experimentation (see Supplementary Figs. 14 and 15 for details), rewardingly, Mitsunobu-mediated chiral resolution of *rac*-**14** was realized with the inexpensive chiral alcohol **13** at 25 °C under DEAD and PPh₃ in PhMe providing the diastereoisomers, but the products were complex because of the poor regioselectivity of the reaction. To address this issue, regioselective protection of the free phenol should be considered. Given the subsequent facile unveiling of the phenol, treatment of *rac*-**14** with BnBr followed by Mitsunobu-mediated chiral resolution (reaction condition: **13**, DEAD, PPh₃ in PhMe), afforded the desired compounds **15** and **16** with *dr* 1.3:1 in an overall 41% yield. After further optimization efforts, gratifyingly, we

**Fig. 3 | Syntheses of (+)−14 and (-)-*ent*−14 by the Mitsunobu-mediated resolution.** DEAD, Diethylazodicarboxylate.

discovered that exposure of *rac*-**14** to BnBr without purification, and subsequent Mitsunobu-mediated chiral resolution (reaction condition: **13**, DEAD, PPh₃ in PhMe/THF) resulted in excellent yield of the desired **15** and **16** with *dr* 1.1:1 (Fig. 3, 2.0 g scale, see Supplementary Pages 30 and 31 for details), which could be readily separated by recrystallization. At this stage, BCl₃-promoted removal of the protective groups in diastereoisomers **15** and **16** readily occurred to deliver (+)- and (-)-**14** in 96% yield with 99% *ee* and 99% *ee* (1.0 g scale), respectively. The absolute configurations of (+)- and (-)-**14** were confirmed by X-ray diffraction analysis (see Supplementary Page 100 for details). The route described above allowed for the facile synthesis of 13 g of (+)- and (-)-**14** (see Supplementary Page 32 for details), which highlights the robust nature of this chemistry. This strategy provides a route to the inaugural asymmetric synthesis of (+)- and (-)-**14** by application of the unique Mitsunobu-mediated chiral resolution on an advanced tricyclic substrate containing the C7 stereocenter.

With the optimized condition in hand, the generality of this strategy was explored using different substrates, including the precursors **11** and **17a** for the synthesis of **2**–**3**, to give various functionalized xanthenes. As shown in Fig. 4a, different 2-substituted and 4-substituted xanthene substrates (*rac*-**8**–**9**, *rac*-**11**, *rac*-**17a**–**17 m**) were tolerated in the optimized condition of the Mitsunobu-mediated chiral resolution to give the corresponding products in good total yields (58%–66%) with excellent *ee* values (92%–99%) in two steps. Hence, this method has broad generality for the asymmetric synthesis of xanthene derivatives. The absolute configuration of (-)-**8** was confirmed by X-ray diffraction analysis (see Supplementary Page 100 for details). Notably, asymmetric syntheses of eight natural products (**8**[53], **9**[29], **11**[54], **17a**[30], **17e**[55], **17 f**[56], **17i**[30], and **17l**[57]) were achieved, overcoming the difficulties previously encountered in the asymmetric synthesis of **11**[48], **17**, and **17e**−**17 f**. Furthermore, the route described above allowed for the facile synthesis of 2.5 g of (+)-**8** and (-)-**8**, 1.3 g of (+)-**11** and (-)-**11** as well as 2.6 g of (+)-**17a** and (-)-**17a** (see Supplementary Pages 23, 28, 35 for details, respectively). Encouraged by these results, this reaction will be beneficial for the asymmetric synthesis of complex natural products such as the myrtucommunins A and B[58] (Fig. 4b), which is currently underway in our laboratory. Of note, there are currently no approaches

available for the direct asymmetric synthesis of the 2-oxabicyclo[3.3.1] nonane scaffold, which is present in biologically interesting complex natural products such as myrtucyclitone C[59]. To our delight, (-)-**17n**, and (-)-**17o**, which is probably the key intermediate for the asymmetric synthesis of myrtucyclitone C, were provided by this Mitsunobu-mediated chiral resolution at an excellent 98.5% *ee* and 97% *ee*, respectively (see Supplementary Pages 73, 74, 76 and 77 for details). The absolute configurations of (-)-**17n** and (-)-**17o** were confirmed by X-ray diffraction (see Supplementary Page 100 for details). These results revealed that the unique Mitsunobu-mediated chiral resolution could be regarded as an alternative strategy for construction of synthetically difficult polycyclic chiral scaffolds.

Having established a reliable method for constructing (+)- and (-)-**14** as well as their congeners, we then turned our focus to the asymmetric synthesis of (+)- and (-)-myrtucommulone D (**1**). Initially, we focused on the synthesis of **Int-2** from **12** and **18** by a Knoevenagel reaction (Fig. 5a). This reaction is challenging not only because of the unfavorable regioselective Knoevenagel reaction of **18** with three carbonyl groups, but also because of the concomitant chemoselective *retro*-Aldol reaction (Fig. 5a, via Path II). Nevertheless, we speculated that the aldehyde group in **14** and the triketone in **12** are a strong nucleophilic acceptor and nucleophile, respectively, which possess high reactivity and would make it possible for this Knoevenagel reaction to proceed. The Rieche reaction of **14** with dichloromethyl ether in CH₂Cl₂ led to aldehyde **18** in 78% yield (1.2 g scale, see Supplementary Page 79 for details). On the basis of Romo's work[60], the Knoevenagel reaction of **12** and **18** under pyrrolidine as the base condition in CH₂Cl₂ proceeded smoothly to provide **14**, but not **Int-2** (via Path I) or **19** (Fig. 5a, via Path I, then Path III). We next turned to a variety of different bases (piperidine, Na₂CO₃, Et₃N, and L-proline) in CH₂Cl₂, which forged the undesired **14** along with low yield (<10%) of the hemiketal **19** and some recovered starting material with no observed **Int-2** under these conditions. To overcome this problem, we hypothesized that it may be possible to differentiate the reaction pathways resulting in **14**, **Int-2**, and **19** by selecting suitable solvents. After an extensive investigation, with L-proline (1 equiv.) as the base and DMF as the solvent, though the desired **Int-2** was not afforded, the cyclization product **19** with *dr* 1:1

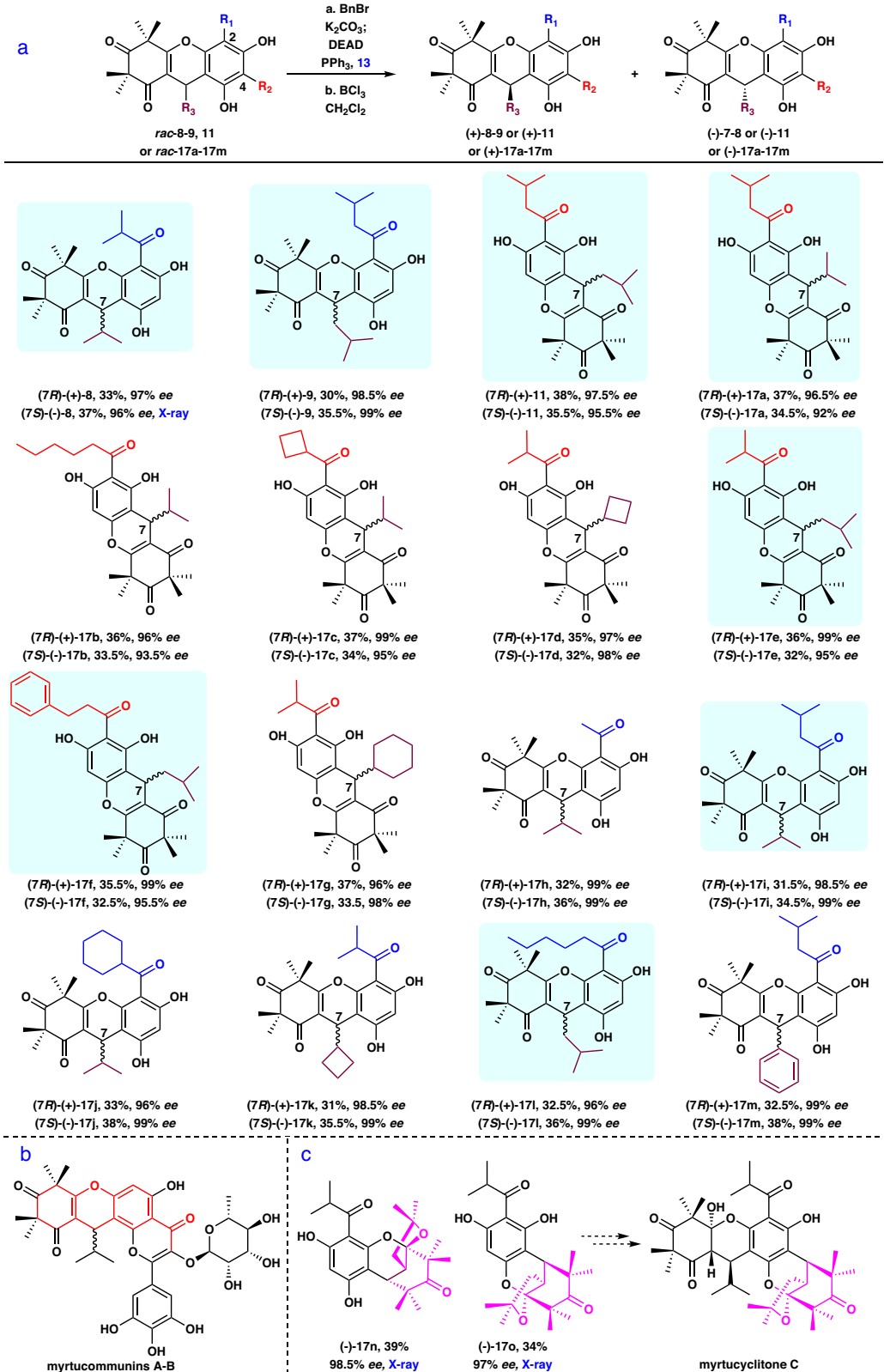

**Fig. 4 | Substrate scope of Mitsunobu-mediated resolution and potential applications. a** Mitsunobu-mediated resolution of *rac*−**8-9**, **11** or *rac*−**17a-17m**. **b** The structures of myrtucommunins A-B with a xanthene scaffold. **c** The structures of highly optically pure (-)−**17n** and (-)−**17o** obtained through Mitsunobu-mediated resolution, along with the potential application of (-)−**17o**. Reaction conditions:[a]*rac*-**7**-**8** (1.0 equiv.), or *rac*-**11** (1.0 equiv.), or *rac*-**17a-17o** (1.0 equiv.),

K₂CO₃ (1.5 equiv.), BnBr (1.05 equiv.), 60 °C, acetone (0.05 M), 10 h, then filtered and concentrated, then DEAD (1.5 equiv.), PPh₃ (1.5 equiv.), **13** (1.2 equiv.), 0-25 °C, THF/PhMe (V/V = 1:1, 0.1 M), 1 h. [b]BCl₃ (10.0 equiv.), CH₂Cl₂ (0.1 M), −50 °C, 1 h. [c]Isolated yields for 2 steps. [d]The *ee* values were determined by chiral HPLC analysis. Blue box with a background refers to natural products.

**Fig. 5 | Asymmetric syntheses of 1, 5, and 21-22. a, b** Asymmetric syntheses of (+) and (-)−**1** in addition to (+)- and (-)−**5**. **c** Asymmetric syntheses of **21** and **22**.

was generated in 86% yield at 25 °C. Subsequently, we envisaged that *retro*-hemiketalization of **19** under strong basic conditions (i.e., LDA, and LiHMDS) provided **Int-2**. To our disappointment, the starting materials were recovered under basic conditions. At this stage, we proposed that *retro*-hemiketalization may occur. However, the above reaction was prone to form the more structurally stable **19** instead of **Int-2** after quenching with solution. Thus, we expected that direct treatment of **19** with the isopropyl reagents would afford **1**. We found that using a cascade reaction of hemiketal **19** to directly construct the

three contiguous stereocenters at C7, C1′, and C6′ posed significant challenging. It was difficult to use hemiketal **19** to provide **1** in the presence of various isopropyl reagents [*i*PrMgBr, *i*PrLi, and (*i*Pr)₂Zn] and different solvents (THF, Et₂O, CH₂Cl₂, and PhMe). Only some starting materials were recovered, probably because of the presence of the competing 1, 2-addition reaction. Subsequently, given the important utility of a cuprous reagent in the Michael addition reaction[61,62], we initially sought to take advantage of this together with the reactivity of **19**. To our disappointment, when we treated **19** and Grignard reagents

or zincon with 3% to 30% of copper catalysts (i.e., CuI, CuBr, or CuBr·SMe₂), the reactions were unsuccessful. At this stage, we envisioned that the complex was formed between Cu$^I$ or Cu$^{II}$ and the carbonyl group at C8 with an oxyanion at C3, which led to undesirable results[63,64]. Gratifyingly, treatment of **19** with *i*PrMgBr (3.5 equiv.) and an excess of CuI (1.1 equiv.) in THF at −50 °C afforded the desired (+)-myrtucommulone D (**1**) in 18% yield. Meanwhile, some starting materials were recovered. Encouraged by this result, we further explored a variety of solvents including CH₂Cl₂, PhMe, Et₂O, 1,4-dioxane and THF/CH₂Cl₂. Rewardingly, using THF/CH₂Cl₂ instead of THF as the solvent accelerated the transformation, increasing the yield of (+)-**1** to 58%. Subsequent screening of copper reagents (CuI, CuBr, CuCN, CuBr·SMe₂) revealed that CuCN provided the most excellent yield for **1**. After extensive investigation, the optimal protocol was identified: when **19** was treated with *i*PrMgBr (3.5 equiv.) in the presence of CuCN (1.1 equiv.) in THF/CH₂Cl₂ at −78 °C to −50 °C, (+)-**1** was obtained in 81% yield without the need for protecting groups. The structure of (+)-**1** was confirmed by X-ray diffraction analysis (see Supplementary Page 99 for details). Subsequently, to further improve the efficiency for the synthesis of (+)-**1**, we explored a sequence of a Knoevenagel-hemiketalization annulation reaction of **12** and **18** without purification followed by *i*PrMgBr and CuCN to give **1** in 66% yield (120 mg scale, see Supplementary Pages 81-82 for details). Notably, a xanthene core and three contiguous stereocenters and one quaternary stereocenter were constructed with high diastereoselectivity by a two-step reaction in one pot. Encouraged by this success, a sequence of Knoevenagel-hemiketalization, and subsequent *i*PrMgBr and CuCN as well as elimination with *p*-TsOH, was exploited to afford (+)-myrtucommulone E (**5**) in 58% yield (120 mg scale, see Supplementary Pages 91 and 92 for details). The structure of (+)-**5** was confirmed by X-ray diffraction analysis (see Supplementary Page 99 for details). Moreover, the preparation of (-)-myrtucommulone D (**1**) and (-)-myrtucommulone E (**5**) were completed from (-)-**14** (Fig. 5b, see Supplementary Pages 81 and 82 and 91 and 92 for details). The ¹H and ¹³C NMR spectra of the newly synthesized **1** and **5** were identical to those of the natural products (see Supplementary Tables 2−3 and 7 for details). Of note, we have also tried to provide **1** and **5** via various other routes (see Supplementary Fig. 4−6 for details, respectively). For example, with **8** in hand, **8** and **20** underwent a Friedel-Crafts type Michael addition (K₂CO₃/CH₂Cl₂) to assemble **21** and **22** with *dr* 1:1.1 in 92% yield (Fig. 5c). However, unfortunately, the regioselective intramolecular cyclization followed by dehydration of **22** with different acids (i.e., *p*-TsOH, CSA, TFA, TiCl₄, and BF₃·Et₂O) did not furnish **5** (see Supplementary Fig. 5 for details). Therefore, the new synthetic route reported in this article was highly efficient for the asymmetric synthesis of **1** and **5** with an angular [6-6-6-6-6] pentacyclic skeleton.

Encouraged by the above experimental outcome, we next turned our attention to the two possible mechanisms responsible for the diastereoselective construction of **1** from **19**, which are proposed in Fig. 6a (Path A, highlighted in red; Path B, highlighted in blue). Subsequently, to gain insight into which pathway is more probable, the density functional theory (DFT) calculations were carried out. We found that *retro*-hemiketalization of **19** (Path A) occurred with a lower barrier than that of the Michael addition of **19** (Path B) to give **Int-5** and **Int-6** (see Supplementary Fig. 80 for details), which implied that hemiketal **19** is more likely to proceed via path A to give **1**. However, the diastereoselective construction of the three contiguous stereocenters remains an interesting problem. Initially, the optimization of the 3D structure of **Int-3a** by DFT was afforded because the anionic **Int-3** could not be found to be a minimum on the potential energy surface, and the steric effects of **Int-3** and **Int-3a** were similar. Clearly, Michael addition of **Int-3** would occur on the *Re*-face (Fig. 6b) leading to the generation of **Int-4** because of less steric hindrance. Subsequently, the thermodynamic process of **Int-4** to **1** was suggested by the following experiments: a. treatment of **1** with NIS and NaOH

in THF at −40 °C to 25 °C resulted in the sprio-**1a** (Fig. 6c, see Supplementary Fig. 75 for details), which indicated that *retro*-hemiketalization of **1** indeed occurred (most likely via **Int-9**); b. exposure of **1** to NaOH and THF without NIS at the same temperature range could only recover the starting material. Then, the subsequent DFT-minimized structures of **Int-7** and **Int-8** were obtained, which showed that an intramolecular oxa-Michael addition of **Int-4** would afford **Int-7** due to the more stable envelope-like conformation in the A ring, with a free energy that was 6.0 kcal/mol lower than that of **Int-8** (Fig. 6a, d). Similarly, comparison of the conformations and free energies of **1** and 1′-*epi*-**1** also showed good diastereoselectivity toward the formation of **1** would be gave (Fig. 6a, d). Thus, the mechanism of stereoselective formation of **1** was explained. Based on the above results, we hypothesized that the structural configurations of (-)-callistenone D (**2**)[32] and (-)-rhodomyrtosone R (**3**) are probably identical, and the structural assignment of (-)-**3** probably needs to be revised. To validate our proposal, we would plan to use the above strategy to achieve (-)−**2** and (-)-**3**.

## Asymmetric syntheses of (+)- and (-)-2−3 as well as (+)- and (-)−6−7

Having achieved the asymmetric syntheses of (+)- and (-)-**1** as well as (+)- and (-)-**5**, and gained an understanding of the mechanism of the cascade reaction, we then set out to the construction of (+)- and (-)-**2**−**3** as well as (+)- and (-)-**6**−**7** as shown in Fig. 7a-c. The preparation of (+)- and (-)-**2**−**3** and (+)- and (-)-**6**−**7** was obtained from (+)-**11**, (-)-**11**, (+)-**17a**, and (-)-**17a**, respectively, through a synthetic route similar to that used to generate **1** and **5** (Fig. 7a, see Supplementary Pages 81 and 91 for details). The structure of (+)-**2** was confirmed by X-ray diffraction analysis (see Supplementary Page 99 for details). The ¹H and ¹³C NMR spectra of the newly synthesized **2**−**3** and **6**−**7** were identical to those of the natural products (see Supplementary Tables 4−6, 8−10 for details, respectively). Notably, among them, the absolute configurations of natural **3** and **7** have not been reported previously[29,31], and the signs of the optical rotation of natural **3** and **7** were identical to those of the newly synthesized (-)-**3** and (-)-**7**. Therefore, the structure of natural **3** was revised to be (1′*S*, 6′*R*, 7 *R*, 7″*S*) (Fig. 7b), and the absolute configuration of natural **7** was determined to be (7 *S*, 7″*S*) (Fig. 7c). These outcomes demonstrate that this strategy has the huge potential for the preparation of more complex compounds with a xanthene core.

## Antibacterial activities of synthetic compounds in vitro and in vivo

Next, we evaluated the antibacterial activity of the 66 compounds prepared in the present study against seven Gram-positive and three Gram-negative bacteria. The results showed that there was no significant difference in the antibacterial activity between the enantiomers. In addition, the hemiketal moiety plays a critical role in their antibacterial activity, based on the comparison of the MICs of **1**-**3** and **5**-**7**. Moreover, **22** exhibited the most potent antibacterial activity against MRSA (MIC 0.5 μg/mL), compared to other synthetic compounds (see Supplementary Table 11 for details). Therefore, we evaluated the antibacterial activity of **22** in vivo. As expected, the treatment of **22** (3.75 mg/kg) resulted in a reduction of wound area and the prevention of skin ulcer formation in a mouse skin infection model (*P* < 0.0001). This effect was comparable to that of vancomycin (see Supplementary Fig. 254 for details). In addition, we compared the time of emergence of resistance in *S. aureus* induced by **22** and norfloxacin. The results showed that the time of induction of drug resistance by **22** was significantly longer than that by norfloxacin (see Supplementary Fig. 255 for details).

## Investigation on the mechanism of action

To elucidate the mode of action of **22**, we successfully obtained four strains (SA₂₂-SR-1 to SA₂₂-SR-4) that were spontaneously resistant to **22**

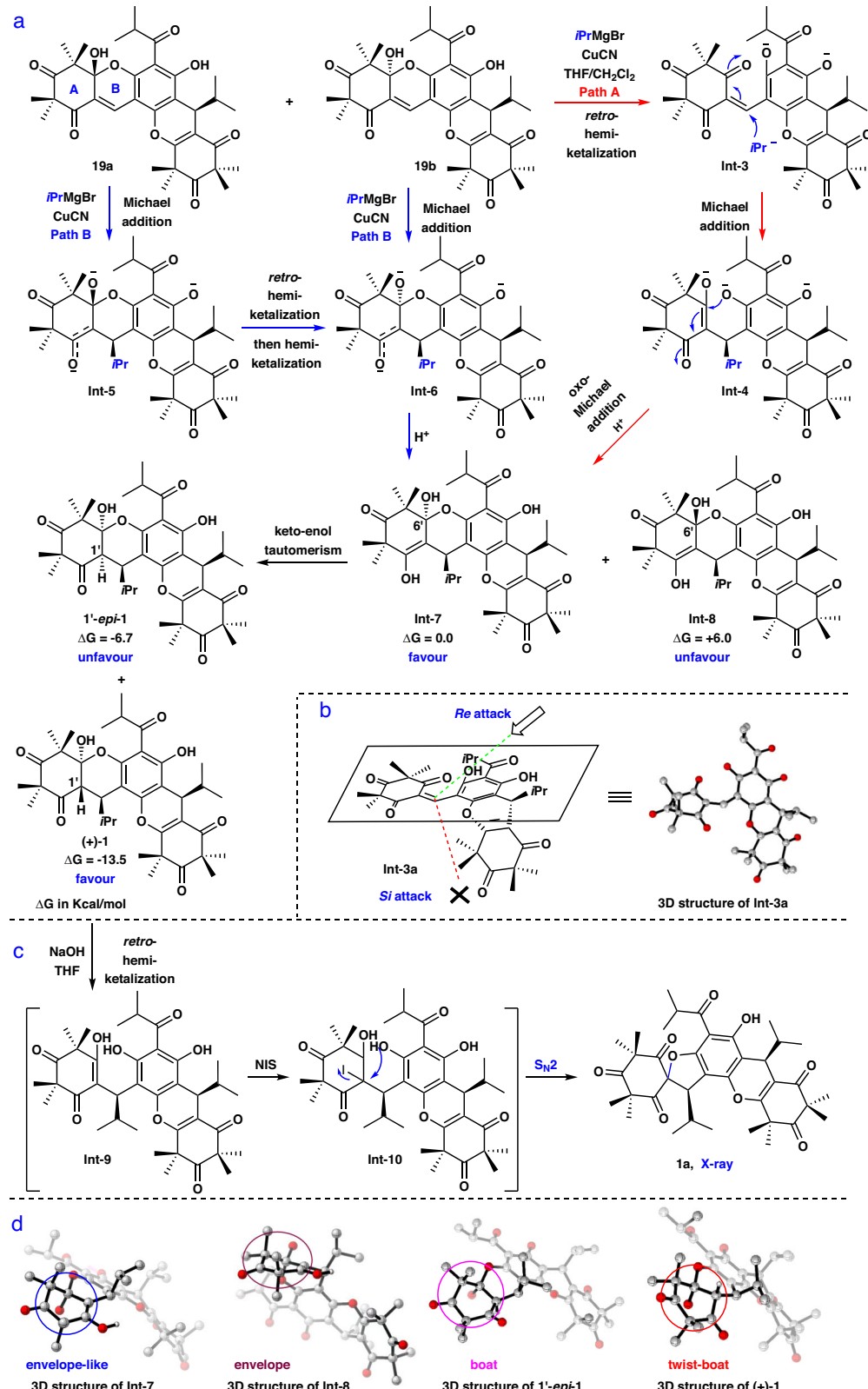

**Fig. 6 | Mechanistic investigation of the diastereoselective synthesis of 1. a** The proposed and computational studies to understand mechanism of diastereoselectivity of the cascade reaction. **b** Depiction of the attack onto the two faces of the Int-3a. **c** Investigation of *retro*-hemiketalization. **d** 3D structures of **Int**-7-8, 1'-*epi*−1 and (+)−1 with their A ring highlighted in four different colored circles. NIS *N*-Iodosuccinimide.

through serial passaging (see Supplementary Table 12 for details). We initially confirmed that the susceptibility of SA$_{22\text{-SR}}$ remained unchanged relative to other antibacterial agents, including vancomycin, ofloxacin, linezolid, kanamycin, meropenem, and tigecycline (see Supplementary Table 13 for details). Then the genomic differences between the wild-type (WT) parental strain SA$_{29213}$ and the spontaneously resistant strains were analyzed. Nine genomic differences, including seven single nucleotide (*gyrA, walK, valS, plsY*, and three

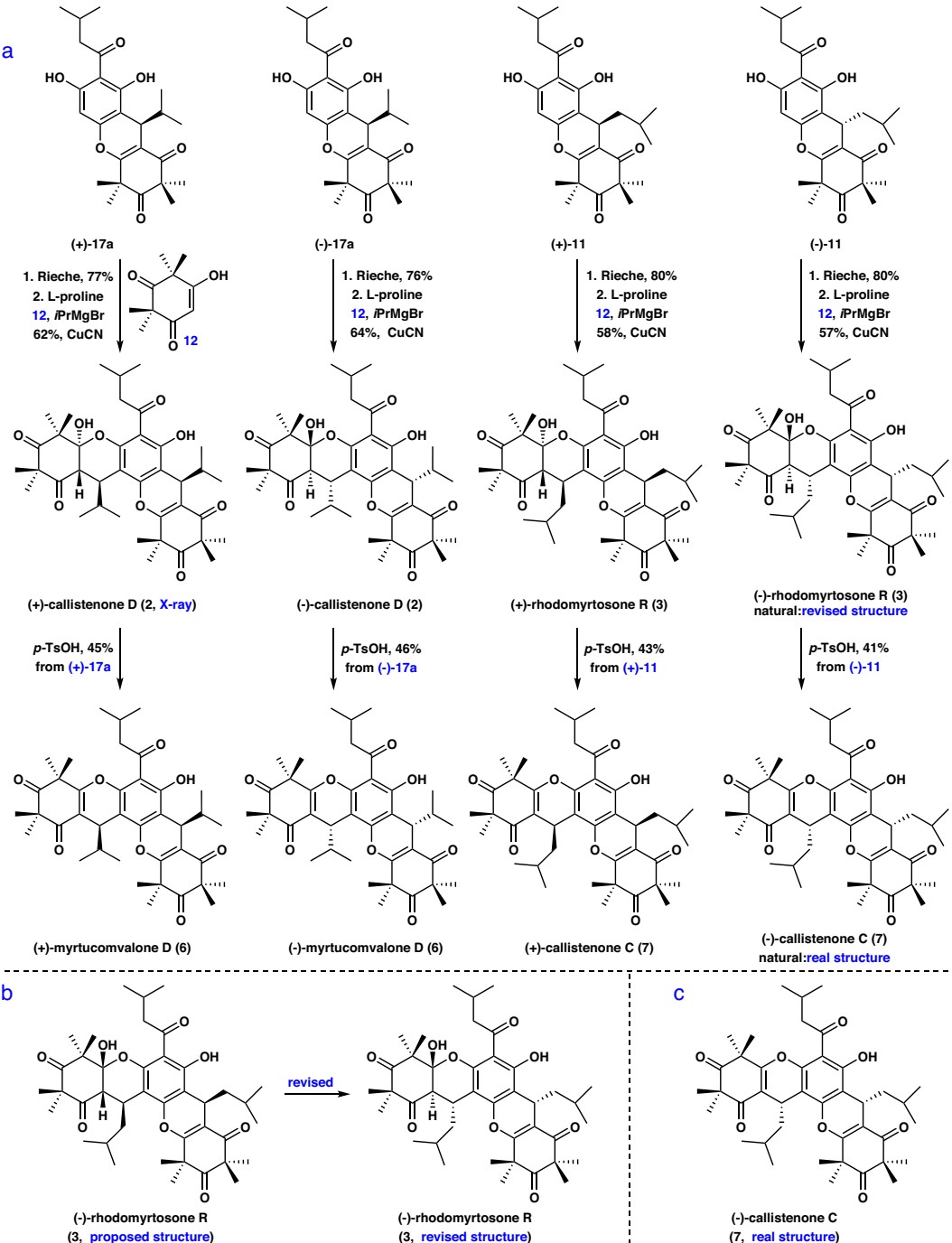

**Fig. 7 | Asymmetric syntheses of 2-3, 6-7 and stereochemical assignment of 3 and 7. a** Asymmetric syntheses of (+)- and (-)-**2**–**3** as well as (+)- and (-)-**6**–**7**. **b**, **c** The structural revisions of natural (-)-**3** and (-)-**7**.

hypothetical genes) and two indels (*glpK* and a hypothetical gene), were identified in all SA$_{22\text{-SR}}$ strains (see Supplementary Table 14 for details). We selected one of the strains with the least number of mutations (SA$_{22\text{-SR-1}}$) for subsequent experiments, so as to exclude the interference of other non-shared mutations (see Supplementary Table 14 for details) as much as possible. To the best of our knowledge, WalK histidine kinase is essential and specific to low G + C Gram-positive bacteria such as *S. aureus*, and has been recognized as promising targets[65]. Given the antibiogram of **22**, which is mainly effective against Gram-positive bacteria, WalK could be considered as a reasonable target. Thus, we focused on examining the effects of **22** on the activity of WalK.

WalK is a transmembrane histidine kinase that works in conjunction with the response regulator WalR to regulate the expression of autolysins, as well as virulence

factors such as hemolysis genes *hla*, *hlb* and *hlgABC*, and genes responsible for biofilm formation in *S. aureus*. In this study, we observed increased transcription levels of WalKR-regulated genes (*atlA*, *lytM*, *hla*, *ssaA*, *sceD*, *SA0710*, *SA2097*, *SA2353*, and *isaA*), increased sensitivity to lysostaphin-induced lysis of SA$_{29213}$, biofilm formation and hemolysis on sheep blood agar after **22** treatment (Fig. 8a–c, see Supplementary Fig. 256 for details), indicating the activation activity of **22** on WalK. In contrast, compared to SA$_{29213}$, SA$_{22\text{-SR-1}}$ exhibited decreased transcription levels of WalKR-regulated

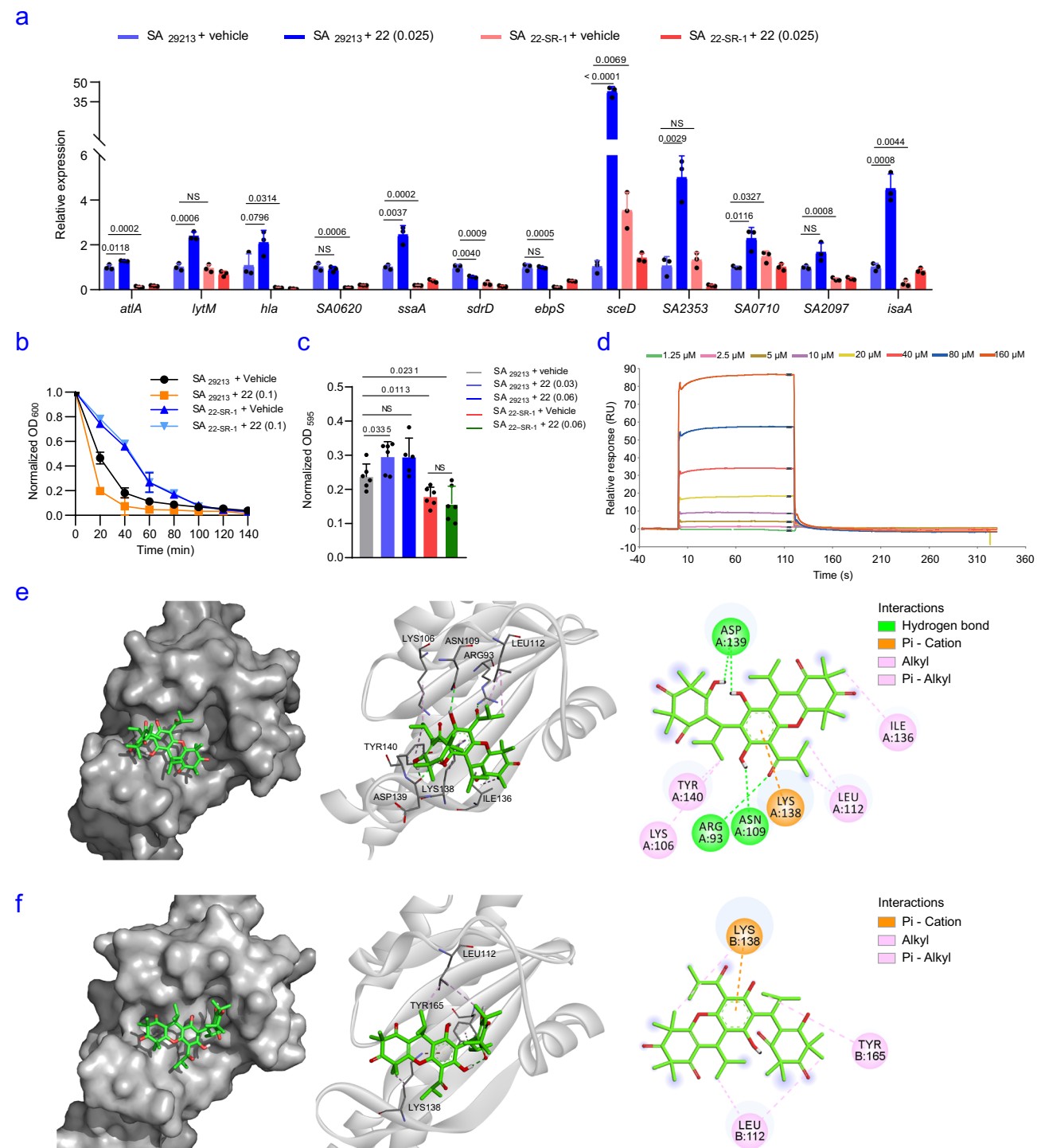

**Fig. 8 | Effects of 22 on WalK activity. a** Transcription levels of genes regulated by WalKR in the presence or absence of **22** (0.1 μg/mL). **b** Lysostaphin-induced lysis process in the presence or absence of **22** (0.025 μg/mL). **c** Biofilm formation of different strains in the presence or absence of **22** (0.03, 0.06 μg/mL). Three biologically independent experiments at least were performed in **a–c**. The mean is shown, and error bars represent the s.d. *P* values were determined using a non-parametric one-way ANOVA. **d** SPR analysis demonstrated the binding of **22** to WalK. **e 22** docked with the erWalK structure (PDB ID: 5IS1). The space-filling model shows the binding of **22** in the β-sheet pocket. **f 22** docked with the erWalK_{R86C} structure (PDB ID: 7DUD). The protein residues, hydrogen bonds and hydrophobic bonds are shown in a solid-ribbon and in 2D view. Pi-Donor hydrogen bonds, conventional hydrogen bonds and hydrophobic bonds are colored orange, green and purple, respectively. Source data are provided as a Source Data file.

genes (*atlA, hla, SA0620, ssaA, sdrD, ebpS, sceD, SA2097, isaA*), biofilm formation, hemolysis on sheep blood agar and inhibited lysostaphin-induced lysis (Fig. 8a–c, see Supplementary Fig. 256 for details), indicating that the decreased activity of WalK in SA_{22-SR-1} which probably due to the R86C mutation in WalK. As expected, all the changes

associated with WalKR function were not affected by **22** in the SA_{22-SR-1} cells (Fig. 8a–c). These results suggested that the WalKR pathway was activated by **22**.

We then employed CRISPR-Cas9 and successfully recreated the R86C mutation [SA_{WalK(R86C)}] in the SA_{29213} background (see Methods

section for details). The MIC of **22** against SA$_{WalK(R86C)}$ was 2 μg/mL, indicating that the level of resistance did not reach that observed in SA$_{22-SR}$ strains. The transcription levels of *lytM, hla, ssaA, sdrD, ebpS, sceD, SA0710, SA2097* and *SA2353* were decreased in SA$_{WalK(R86C)}$, indicating that the function of WalK was down-regulated (see Supplementary Fig. 257 for details). The MIC of vancomycin against SA$_{WalK(R86C)}$ was 2 μg/mL, suggesting that other point mutations in SA$_{22-SR}$ strains contribute to restoring vancomycin susceptibility. Our results, along with previous reports that have generated isogenic mutants[66–68], suggest that mutations at different positions of WalK contribute differently to vancomycin susceptibility. However, we found that the R86C mutation of WalK is highly unstable and usually spontaneous reverses to the wild state after 2-3 passages of culture. We deduced that some of other point mutations in SA$_{22-SR}$ strains may act as necessary compensation for the *walK* mutation. Therefore, we constructed SA$_{22-SR-1:walK}$ by the episomal expression of wild-type *walK* in SA$_{22-SR-1}$. Our results showed that transcription level of genes which are positively regulated by WalKR and the sensitivity to lysostaphin-induced lysis in the presence of **22** were restored in SA$_{22-SR-1:walK}$ cells, suggesting that **22** may act on wild type WalK (see Supplementary Fig. 258 for details).

For more evidence, the effect of **22** on bacterial autolysis was investigated by examining the membrane integrity. As expected, we observed that the use of **22** significantly increased the uptake of propidium iodide (PI) (see Supplementary Fig. 259a for details) and increased numbers of dead bacteria as the cell membranes burst (see Supplementary Fig. 259b for details). Moreover, using the fluorescence agent DiSC$_3$(5), we observed a significant decrease in fluorescence intensity after treatment of **22**, suggesting that this compound caused hyperpolarization of the bacterial membrane (see Supplementary Fig. 259c for details).

To verify this speculation, we turned attention to exploring the binding between **22** and WalK. Since the full-length protein is prone to degradation, we purified the extracellular part of the receptor WalK (erWalK, residues 35-182 of uniprot ID: Q6GKS6) (see Supplementary Fig. 260-262 for details) and determined the binding kinetics of **22** to erWalK using surface plasmon resonance (SPR) analysis. The results showed an association rate constant $k_a$ of $9.32 \times 10^2 \, M^{-1}s^{-1}$, a dissociation rate constant $k_d$ of $8.45 \times 10^{-3} \, s^{-1}$, and an equilibrium dissociation constant $K_D$ of $1.11 \times 10^{-5} \, M$ (Fig. 8d), indicating that **22** can directly interact with WalK. As expected, the SPR result of **22** and erWalK$_{R86C}$ (erWalK with R86C mutation) showed an equilibrium dissociation constant $K_D$ of $7.38 \times 10^{-5} \, M$, indicating a marked decrease in the affinity of erWalK$_{R86C}$ with **22** compared to erWalK (see Supplementary Fig. 263 for details).

To further investigate this interaction, we docked **22** into the wild-type structure of WalK using Autodock Vina. The ligand was found to interact with WalK in the α1 and β2 regions where the mutant structure differs most from the wild-type structure. Specifically, **22** forms three hydrogen bonds within the β-sheet pocket formed by β1-5 and α1 (Fig. 8e, see Supplementary Fig. 264 for details). The calculated affinity of **22** for erWalK using AutoDock Tools is $2.01 \times 10^{-5} \, M$, which is consistent with the results obtained using SPR. The three residues forming hydrogen bonds with **22** are Arg93, Asp139 and Asn109, and the hydrocarbon potion of Lys106, and residues Leu112, Ile136, and Tyr140 interacts with **22** by hydrophobic interactions (Fig. 8e, Supplementary Table 17 for details). In contrast, **22** docked poorly a predicted affinity of $6.56 \times 10^{-5} \, M$ to the same pocket in erWalK$_{R86C}$ (PDB ID: 7DUD, Supplementary Table 16 for details), which was obtained in our study. The local structural rearrangements in R86C diminished most of the interactions, leaving only Lys138 for pi-cation interaction and Leu112 and Try165 for hydrophobic interactions (Fig. 8f, Supplementary Table 18 for details). Taken together, these results indicated that **22** activates the WalK function. To our knowledge, WalK inhibitor has been reported[69], whereas its activators have not been reported. The present results confirm that the WalKR pathway is a vulnerable site in *Staphylococcus*, and **22** represents the inaugural WalK activator with antibacterial activity.

Of note, due to the differences in structure between the erWalK of *Streptococcus pneumoniae* (*S. pneumoniae*; extra cellular PAS domain only has one transmembrane helix) and *S. aureus*, as well as the MIC for *S. pneumoniae* with compound **22** is only double that of *S. aureus*, it would suggest that **22** may not be specific for erWalK and could potentially have a second target. The variance in resistance levels between SA$_{WalK(R86C)}$ and SA$_{22-SR}$ strains to **22** further supports this hypothesis. Therefore, we must point out that although this study identified the interaction between **22** and WalK, it cannot exclude the possibility of other targets. Other essential gene mutations in SA$_{22-SR}$, such as *plsY*, which are worthy of further investigation.

In this work, we present a concise approach for the collective asymmetric total syntheses of (+) and (-)-myrtucommulone D (**1**), (+) and (-)-callistenone D (**2**), (+) and (-)-rhodomyrtosone R (**3**), (+) and (-)-myrtucommulone E (**5**), (+) and (-)-myrtucomvalone D (**6**), and (+) and (-)-callistenone C (**7**) in six steps[70–72]. The key to the synthesis is the use of a Mitsunobu-mediated chiral resolution with broad substrate applicability and high optical purity (92%–99% *ee*). Additionally, the establishment of a unique Knoevenagel/hemiketalization annulation followed by a *retro*-hemiketalization/double Michael addition cascade reaction not only constructs the [6-6-6-6-6] pentacyclic system, but also stereoselectively introduces three consecutive stereocenters and a quaternary stereocenter. The mechanism of this stereoselective transformation is illustrated by Quantum mechanical calculations. Based on the total synthesis, the absolute configurations of rhodomyrtosone R and callistenone C are determined. Notably, this synthesis provides 66 compounds for further study of their antibacterial activity. Among them, compound **22** shows the most effective activity against MRSA in vitro and in vivo. Genetic and biochemical studies revealed that **22** is an antibacterial agent to affect autolysis by activating WalK, making it a promising lead compound with an unusual mechanism that is unlikely to lead to drug resistance[73]. This work lays the foundation for the asymmetric synthesis of various polycyclic xanthenes and will accelerate the development of antibacterial agents.

## Methods
### General information on synthetic experiment
Unless otherwise mentioned, all reactions were carried out under a nitrogen atmosphere under anhydrous conditions and all reagents were purchased from commercial suppliers without further purification. Solvent purification was conducted according to *Purification of Laboratory Chemicals* (Peerrin, D. D.; Armarego, W. L. and Perrins, D. R., Pergamon Press: Oxford, 1980). Yields refer to chromatographically and spectroscopically ($^1$H NMR) homogeneous materials, unless otherwise stated. Reactions were monitored by thin layer chromatography on plates (GF254) supplied by Yantai Chemicals (China) using UV light as visualizing agent, an ethanolic solution of phosphomolybdic acid, or basic aqueous potassium permanganate (KMnO$_4$), and heat as developing agents. If not specially mentioned, flash column chromatography uses silica gel (200-300 mesh) supplied by Tsingtao Haiyang Chemicals (China), preparative thin layer chromatography (PTLC) separations were carried out 0.50 mm Yantai (China) silica gel plates. NMR spectra was recorded on Bruker AV600, AV500, Bruker ARX400, and calibrated using residual undeuterated solvent as an internal reference (CHCl$_3$, δ 7.26 ppm $^1$H NMR, δ 77.20 $^{13}$C NMR). The following abbreviations were used to explain the multiplicities: s = singlet, d = doublet, t = triplet, q = quartet, b = broad, m = multiplet. High-resolution mass spectra (HRMS) was recorded on a Bruker Apex IV FTMS mass spectrometer using ESI (electrospray ionization). Infrared spectra was recorded on a Shimadzu IR Prestige 21, using thin films of the sample on KBr plates. Optical rotations were measured with a

Rudolph autopol I automatic polarimeter using 10 cm glass cells with a sodium 589 nm filter.

## Antibacterial activity experiment

**Microorganisms.** *Staphylococcus aureus* ATCC 29213 [methicillin-susceptible *S. aureus* (MSSA)], *S. aureus* CC 48973 [methicillin-resistant *S. aureus* (MRSA)], *S. aureus* CC 49050 [methicillin-Sensitive *S. aureus* (MSSA)], *Enterococcus faecalis* (*E. faecalis*) ATCC 29212, *Enterococcus faecium* CC 42266 [vancomycin-resistant *E. faecium* (VRE)], *Streptococcus pneumoniae* CC F3993 [penicillin susceptible *S. pneumoniae* (PSSP)], *Klebsiella pneumoniae* (*K. pneumoniae*) ATCC 13883, *Pseudomonas aeruginosa* (*P. aeruginosa*) ATCC 27853, *Escherichia coli* (*E. coli*) ATCC 25922 were standard isolates obtained from ATCC (American Tissue Culture and Collection, Manassas, VA, USA), and CC (Clinical Collection), respectively. Of note, the genome sequences of the above mentioned clinical strains have been deposited in the National Center for Biotechnology Information under the common deposition number PRJNA1054306.

**Antimicrobial agents and medium.** Four antibacterial agents, including daptomycin, vancomycin, polymyxin B and oxacillin, were purchased from National Institutes for Food and Drug Control, People's Republic of China. Cation-adjusted Mueller-Hinton (CAMH) broth was purchased from BD (Cockeysville, MD) and prepared according to the recommendations of the Clinical and Laboratory Standards Institute (CLSI, formerly NCCLS)[74].

Stock solutions of antibiotics were prepared in solvents and diluents recommended by CLSI based on their actual purity or potency. Before use, these solutions were sterilized through $0.22\,\mu m$ filters. Test solutions with different concentrations of compounds and the antimicrobials were obtained by two-fold serial dilutions with CAMH broth.

Trypticase soy broth (TSB; BD, Cockeysville, MD) was used for bacterial growth, while CAMH broth was employed for all susceptibility testing. Colony counts were determined using tryptic soy agar (TSA; BD, Cockeysville, MD) plates.

**Antibacterial activities of synthetic compounds in vitro.** The MICs of the antibacterial agents for all strains were determined using the broth microdilution method according to CLSI guidelines[75]. Wells of 96-well microtiter plates (Nunc, Thermo Fisher Scientific Inc., Roskilde) were inoculated with $100\,\mu L$ of CAMH broth or CAMH broth with 3% lysed horse blood (*S. pneumoniae* only), containing serially-diluted antimicrobials and a final inoculum of $5 \times 10^5$ CFU/mL. The concentration ranges were from 0.0625 to $128\,\mu g/mL$ for each compound: daptomycin, vancomycin, polymyxin B and oxacillin. The microtiter plates were incubated at $37\,°C$ for 24 h. The MIC was defined as the lowest concentration of an antimicrobial agent that inhibited 90% of bacterial growth determined by the MTT assay (MCE, USA)]. All MIC determinations were performed in duplicate.

**Efficacy of 22 in vivo.** 6–8-week-old female Balb/c mice weighing $20 \pm 2$ g were purchased from Beijing HFK Bioscience Limited Company (Beijing, China). All animals were housed in a specific-pathogen-free environment, with housing conditions including dark/light cycle 12/12, an ambient temperature around 21–22 °C and humidity between 40 and 70% (average of 55%). For the wound healing experiment, animals were grouped randomly into 4 groups (n = 3 per group). For the bacterial loading experiment, animals were grouped randomly into 3 groups (n = 20 per group, n = 5 per time point). The dorsal hair of all mice was removed using an electric razor. Subsequently, full-thickness wounds (1.0 cm × 1.0 cm) were created on the back skin of all mice. After 1 h, except for the control, all mice were infected with $50\,\mu L$ of MRSA 252 at a concentration of $2 \times 10^9$ CFU/mL. After 24 h, DMSO was used as a solvent in a total volume of $20\,\mu L$ with a final concentration of

3.75 mg/kg **22,** or vancomycin was dripped onto the wound area twice per day for 7 days. At 1, 3, 5 and 7 d after wounding, each mouse's wound was separately scraped ten times with a sterile cotton swab. The swabs were then placed in 1 mL of sterile saline and agitated to release the bacteria into the liquor. A volume of $5\,\mu L$ of samples after 20-fold series dilution (20, 400, 8,000, 16,000 and 320,000-fold) with sterile saline, was applied to TSA MHA plates and incubated at $37\,°C$ for 24 h. The quantity of bacteria on every plate was calculated as the CFUs with an autocolony counter (Shineso Science & Technology, Hangzhou, China). Plates with either too many colonies (> 300) or too few colonies (<30) were not counted in the results. The progress of wound healing was assessed by an observer who was blinded to the experiments. The open wound area was documented with a digital camera and analysed by using Image J software on day 7. The animal experiments in this study were approved by the 'Animal Ethical and Experimental Committee of the Army Military Medical University (Chongqing, Permit Number 2011-04)' in accordance with their rules and regulations.

**Drug resistance study.** The overnight cultures of SA$_{29213}$ were diluted 1:100 into fresh TSB with sublethal concentrations of compound **22** or norfloxacin (0.5 × MIC). After incubation at $37\,°C$ for 24 h, the MIC value was determined. The process was repeated for 16 passages, with all measurements carried out using biological replicates.

## Investigation on the mechanism of action

**Selection of spontaneous resistant mutants.** Spontaneous mutants resistant to **22** were selected under selective pressure of **22** to investigate its mechanism of action. Serial passaging of SA$_{29213}$ was performed to select mutants spontaneously resistant to **22**. An aliquot of $30\,\mu L$ of bacterial culture at mid-log phase (OD$_{600}$ = 0.6) was added to 3 mL fresh CAMH broth containing 0.2 μg/mL of **22** and subsequently incubated at 37 °C with shaking (200 x g) overnight. The culture was then transferred to fresh medium containing 0.4 μg/mL of **22** at a ratio of 1:100. This serial increase process was repeated until growth was no longer observed. The cells from the highest concentration that supported growth were spread onto TSA agar plates containing 4 μg/mL of **22**. Colonies originating from different tubes represented independent biological events. The resistance phenotype was confirmed by testing for a shift in MIC values.

We obtained four such mutants (SA$_{22\text{-SR}}$) from independent cultures with bacterial growth in TSB containing 10 × MIC (4 μg/mL) (Supplementary Table 12) through serial passaging. The susceptibility of SA$_{22\text{-SR}}$ remained unchanged relative to other antibacterial agents, including vancomycin, ofloxacin, linezolid, kanamycin, meropenem, and tigecycline (Supplementary Table 13).

**Whole-genome sequencing (WGS).** Under normal atmospheric conditions, the cells of SA$_{29213}$ or SA$_{22\text{-SR}}$ were grown in 3 mL TSB to the mid-log phase (OD$_{600}$ = 0.8). A 1 mL culture was centrifuged (10,000 x g, 3 min). The supernatant was discarded and the cells were resuspended in 1 mL of lysostaphin at a final concentration of 5 μg/mL at 37 °C for 30 min. Genomic DNA was extracted from each isolate using the Gram-positive Bacterial Genome Extraction Kit (Shanghai Sangon Biotech).Whole-genome fragment libraries were prepared using the Paired-End Sample Preparation Kit (Illumina). The genomes were sequenced using the Illumina HiSeq 2500 platform (Illumina, San Diego, CA, USA) and assembled with de novo SPAdes Genome Assembler (version 3.12.0)[76]. The resulting reads were mapped to a SA$_{29213}$ reference genome, and mutations were identified using Snippy (https://github.com/tseemann/snippy). The raw reads of SA$_{29213}$ and SA$_{22\text{-SR}}$ strains were deposited in GenBank under BioProject PRJNA1052989.

**Hemolysis activity assay.** $1\,\mu L$ overnight cultures of SA$_{29213}$ in the presence or absence of **22** (0.1 μg/mL) and SA$_{22\text{-SR-1}}$ grown in TSB were

loaded onto a sheep blood plate. The plate was incubated at 37 °C for 24 h and further stored at 4 °C for 24 h before being photographed.

### Gene overexpression and quantitative real-time PCR (qRT-PCR)

**Gene overexpression.** For the gene overexpression study, wild-type *walK* was synthesized using the PCR-based Accurate Synthesis. Restriction endonuclease recognition sites for EcoRV were introduced to facilitate the insertion of the DNA into the expression vector pYJ335. The recombinant plasmids pYJ335-*walK* was then transformed into *E. Coli* TOP 10 cells for cloning and plasmid preparation. The selected plasmid pYJ335-*walK* was further transformed via electroporation into *S. aureus* RN4220 and subsequently into SA$_{22\text{-SR}}$. PCR assays were utilized to verify the successful transformation of the pYJ335-walK plasmid. All primers used in this study are listed in Supplementary Table 15. Anhydrotetracycline (100 ng/mL) was used to induce the gene transcription in SA$_{22\text{-SR-1}:walK}$.

**Quantitative real-time PCR (qRT-PCR).** SA$_{29213}$, SA$_{22\text{-SR-1}}$, SA$_{\text{WalK(R86C)}}$ or SA$_{22\text{-SR-1}:walK}$ cells were grown in 3 mL TSB in the presence or absence of 0.025 μg/mL 22 at 37 °C with shaking (200 x g) until reaching the mid-log phase (OD$_{600}$ = 0.8). An equal volume (30 μL) of DMSO was used as a vehicle control. Anhydrotetracycline (100 ng/mL) was used to induce the gene transcription of SA$_{22\text{-SR-1}:walK}$. A 1 mL culture was centrifuged (10,000 x g, 3 min), and the supernatant was discarded. The cells were resuspended using 1 mL of lysostaphin at a final concentration of 5 μg/mL at 37 °C for 30 min. Total RNA was extracted using the Bacterial Total RNA extraction kit (TIANGEN, Beijing, China). The purified RNA (2 μg) was used for reverse transcription to obtain cDNA with RT Master Mix (MCE, United States). qRT-PCR was performed using SuperScript III Platinum SYBR Green One-Step qRT-PCR with 16 S rRNA as the internal control. The $2^{-\Delta\Delta Ct}$ method was used to determine the fold-changes of gene expression.

**CRISPR-Cas9 gene editing.** Engineered CRISPR/Cas9 System was used to generate a point mutation in the *walK* gene (C256T) according to the previous study[77]. Briefly, spacer 1 (5-GAATTTCTCCAA TTTCTTGA-3) and spacer 2 (5-AAACTCAAGAAATTGGAGAAATTC-3) were synthesized and then phosphorylated in a total volume of 50 μL, which included 2 μL of spacer 1 (50 μM), 2 μL of spacer 2 (50 μM), 5 μL of 10 x T4 DNA ligase buffer (NEB), 1 μL of T4 polynucleotide kinase (Takara) and 40 μL of ddH$_2$O. After incubated at 37 °C for 1 h, 2.5 μL of 1 M NaCl was added and incubated at 95 °C for 3 min. A golden gate assembly experiment was performed in a total volume of 10 μL of system, which included 20 fmol of pCasSA plasmid, 100 fmol of spacers, 1 μL of 10 x T4 DNA ligase buffer (NEB), 0.5 μL of T4 DNA ligase (NEB), 0.5 μL of BsaI-HF (NEB) and ddH$_2$O to 10 μL. After 25 cycles (37 °C for 2 min, 16 °C for 5 min), 50 °C for 5 min and 80 °C for 15 min treatment, the 10 μL product of Golden Gate assembly was transformed into 100 μL *E.coli* DH10B competent cells. Successful colonies were selected on a Luria Broth (LB) agar plate containing 50 μg/mL kanamycin. The success of constructing the pCasSA-spacer plasmid was verified by PCR. The constructed plasmid was then digested with XbaI and XhoI in a total volume of 50 μL, which included 5 μL of 10 x Cutsmart buffer (NEB), 2 μg of plasmid pCasSA-spacer, 2 μL of XbaI (NEB), 2 μL of XhoI (NEB) and ddH$_2$O to 50 μL. After incubated at 37 °C for 8 h, the digested plasmid was purified using the SanPrep PCR purification kit (Sangon Biotech, Shanghai, China).

The upstream and downstream regions of target point mutation, which contain the *walK* (C256T), were amplified by PCR. A Gibson assembly experiment was then performed in a total volume of 20 μL, which included 20 fmol of XbaI/XhoI digested pCasSA-spacer plasmid, 20 fmol of upstream and downstream of *walK* (C256T) and ddH$_2$O to 20 μL. After incubated at 50 °C for 1 h, the Gibson assembly product was transformed into 100 μL DH10B competent cells. Successful colonies were selected on a LB agar plate containing 50 μg/mL

kanamycin. The success of constructing the plasmid pCasSA-WalK was verified by PCR.

The pCasSA-WalK plasmid was transformed into the IM08B strain using electroporation. Colonies were selected on a LB agar plate containing 50 μg/mL kanamycin. The plasmid from the IM08B strain was then transformed into the SA$_{29213}$ strain by electroporation. The transformed cells were plated on a TSB agar plate containing 10 μg/mL chloramphenicol. After overnight incubated at 30 °C, colonies were grown in TSB containing 10 μg/mL chloramphenicol for genomic DNA extraction. The efficiency of genome editing was verified by PCR using the genomic DNA as the PCR template.

The confirmed *S. aureus* mutant containing the pCasSA plasmid was first incubated in TSB at 30 °C overnight. The culture was diluted 1:1000 in 3 mL of TSB and incubated at 42 °C for 18 h. Subsequently, 10 μl of the culture was streaked onto a TSB agar plate in the presence or absence of chloramphenicol (5 μg/mL) and incubated at 37 °C overnight for plasmid curing. All the primers used in this study are listed in Supplementary Table 15 (BioProject number of constructed WalK$_{R86C}$ mutant and spontaneous revertant strains: PRJNA1113534 and PRJNA1113431).

### Lysostaphin-induced lysis, biofilm production assay of 22

**Lysostaphin-induced lysis assay.** The overnight cultures of SA$_{29213}$, SA$_{22\text{-SR-1}}$ or SA$_{22\text{-SR-1}:walK}$ were diluted 1:100 into fresh TSB supplemented with anhydrotetracycline (100 ng/mL) for SA$_{22\text{-SR-1}:walK}$. The cells were grown to an OD$_{600}$ of 1.2 to 1.7, harvested and resuspended in PBS to an OD$_{600}$ of 1.0 with 200 ng/mL lysostaphin supplement in the presence or absence of 0.1 μg/mL **22**. The suspensions were then incubated at 37 °C with shaking (200 x g), and the OD$_{600}$ was monitored at different time points. The measured OD$_{600}$ was normalized by dividing the initial OD$_{600}$ for presentation.

**Biofilm production assay.** The overnight cultures of different *S. aureus* strains were diluted 1:100 into fresh TSB containing 0.5% glucose and 3% NaCl in the presence or absence of **22**. After the cultures were grown in a polystyrene 96-well plate at 37 °C for 24 h, the wells were washed with PBS 3 times and subsequently fixed with 99% methanol for 15 min. The plates were then dried at 37 °C and stained with 1% crystal violet for 8 min. Subsequently, the wells were gently washed with tap water until no color was observed and dried at 37 °C. Acetic acid (33%) was utilized to elute the stained biomass, which was quantified by UV-vis at 595 nm.

### Membrane integrity and polarization, and bacterial viability assay of 22

**Membrane integrity assay.** SA$_{29213}$ cells were washed and resuspended in 0.01 M saline at an OD$_{600}$ of 0.5 Propidium iodide (PI) (Thermo Fisher Scientific, catalog no P1304MP) at a concentration of 10 nM was added in the presence or absence of **22**. After incubation at 37 °C for 30 min, fluorescence was measured with an excitation wavelength of 535 nm and an emission wavelength of 615 nm.

**Membrane polarization assay.** The exponential phase (OD$_{600}$ = 0.5) of SA$_{29213}$ bacterial cells in the presence or absence of **22** was washed and resuspended in 5 mM HEPES buffer containing 5 mM glucose at pH 7.0. 3,3-Dipropylthiadicarbocyanine iodide DiSC$_3$(5) (Aladdin, catalog no. D131315; 0.5 μM) was added, and after 30 min, final concentration of **22** (0, 0.2, 0.4, 0.8, 1.6, 3.2, 6.4 μg/mL) was injected. The membrane potential of bacteria was measured with an interval of 5 min for 60 min, using an excitation wavelength of 622 nm and an emission wavelength of 670 nm with an Infinite M200 microplate reader (Tecan).

**Bacterial viability assay.** A LIVE/DEAD BacLight bacterial viability kit (Invitrogen, catalog no. L7007) was used to evaluate the effect of **22** on

bacterial viability. SA$_{29213}$ cells were cultured overnight, washed three times and resuspended in 0.01 M saline at OD$_{600}$ = 0.5. The bacteria were harvested after overnight incubation in the presence of different concentrations of **22** (0, 0.4 and 1.6 μg/mL) and later washed and resuspended in saline. SYTO9 (1.67 M, 0.7 μL) and PI (10 M, 2.3 μL) were added to each sample with the final volume of 1 mL, and incubated at room temperature in the dark for 15 min. A confocal laser scanning microscope (Leica TCS SP8) was used to obtain fluorescent images of stained bacteria.

**Protein expression and purification.** The cDNA coding for exspectrallular receptor of the WalK (erWalK, 35-182 amino acids) and erWalK$_{R86C}$ were chemically synthesized with optimization for expression in *E. coli* using the PCR-based Accurate Synthesis. Restriction endonuclease recognition sites for NdeI-XhoI were introduced to insert the DNA into the expression vector pET-30a(+). The recombinant plasmids pET-30a(+)-erWalK and pET-30a(+)-erWalK$_{R86C}$ were transformed into *E. Coli* TOP 10 cells for cloning and plasmid preparation, and *E. Coli* BL-21 cells for protein expression, respectively. The BL-21 strain carrying the aforementioned plasmid was grown in LB medium to an OD$_{600}$ of 0.4 at 37 °C. The cells were then supplemented with 1 mM isopropyl β-D-1-thiogalactopyranoside (IPTG). The induced cells were further grown at 37 °C for 4 h.

To purify the recombinant proteins, the cells were harvested and lysed by ultrasonication. The supernatant of the lysed cells was loaded onto Ni-NTA columns (Qiagen, Germany), and the proteins were purified by a gel-filtration column (GE Healthcare, USA) with gel-553 filtration buffer (100 mM NaCl, 10 mM Tris-HCl, pH 7.5 and 1 mM DTT). The protein concentration was determined using the Bradford method.

**Structure determination of erWalK$_{R86C}$**
**Protein crystallization.** Gel filtration-purified proteins (20 mg) were mixed with TEV protease (1 mg/mL, 500 μL) individually and incubated overnight at 4 °C to cleave polyhistidine tags. erWalK$_{R86C}$ was concentrated to 25 mg/mL using a 30-kDa Ultra-15 concentrator (Amicon). Crystal screens were set up for crystallization using hanging drop vapor diffusion. Drops were prepared by mixing the protein solution (10 mM Tris pH 8.0, 150 mM NaCl, 1 mM DTT) with an equal amount of crystallization reservoir solution. The trays were incubated at 18 °C, and crystals were observed after 96 h. The best diffracting APO crystals were grown using a crystallization reservoir solution comprising 0.2 mM Ammonium citrate, and 18% w/v PEG 3350. Crystals were rapidly soaked in reservoir solution supplemented with 20% glycerol as a cryoprotectant, mounted on loops, and flash-cooled at 100 K in a nitrogen gas cryostream. 4.6.2 Crystal diffraction data were collected from a single crystal at the Shanghai Synchrotron Radiation Facility (SSRF) BL18U beamline, China, with a wavelength of 0.9795 Å at 100 K. The diffraction data were processed and scaled with XDS[17]. Relevant statistics and the X-ray diffraction data are summarized Supplementary Table 16.

**Structure determination and refinement.** The structure was solved by the molecular replacement method, with erWalK (PDB ID: 5IS1) used as the starting model, which shares 99% similarity with the erWalK$_{R86C}$ sequence[78]. The initial model was build using PHENIX.autobuild[79]. Manual adjustment of the model was carried out using the proGram COOT, followed by refinement using PHENIX.refine and Refmac5[80–82]. The stereochemical quality of the structures was checked by using PROCHECK[83]. All residues were located in the favored and allowed regions, with none found in the disallowed region, as detailed in the Ramachandran plot. Refinement resulted in a model with excellent refinement statistics and geometry (Supplementary Table 16). Structure validation was performed using the Protein Data Bank ADIT Servers. All structure images were prepared using the PyMOL

Molecular Graphics System (Schrödinger, LLC)[84], UCSF Chimaera[85], and BIOVIA Discovery Studio[86].

**Surface plasmon resonance (SPR) experiment of 22.** To analyze the binding of **22** to erWalK and erWalK$_{R86C}$, SPR experiments were performed using a Biacore 8 K High-throughput Intermolecular Interaction Analysis System. The binding assay was performed utilized a CM5 sensor chip for high-affinity capture of the protein. **22** was diluted in running buffer PBS. Analytes were injected through reference and active channels at a flow rate of 20 μl/min. The association and dissociation times were set to 240 and 160 s, respectively. Affinity fitting was carried out with Biacore evaluation software T200 (GE, United States).

To characterize the binding between **22** and WalK, we purified the exspectrallular part of the receptor WalK (erWalK, residues 35–182 of uniprot ID: Q6GKS6). Binding kinetics of **22** to erWalK was determined by surface plasmon resonance (SPR) analysis.

**Molecular docking of between 22 and WalK or erWalK$_{R86C}$.** The erWalK structure (PDB ID: 5IS1)[78] and erWalK$_{R86C}$ structure (PDB ID: 7DUD) were employed for docking studies. PyMoL (version 1.7.6; https://pymol.org) produced the least energy conformations using default limits. The analysis of protein-ligand docking with ligand binding flexibility and binding pocket residues was performed using AutoDock Vina (version 1.1.2; http://vina.scripps.edu/). Finally, images were generated from PyMoL and BIOVIA Discovery Studio (version 16.1.0, https://www.3ds.com/).

**Reporting summary**
Further information on research design is available in the Nature Portfolio Reporting Summary linked to this article.

## Data availability
The X-ray crystallographic coordinates for structures reported in this study have been deposited at the Cambridge Crystallographic Data Centre (CCDC), under deposition numbers 2248854 (**1**), 2248859 (**1a**), 2248856 (**2**), 2131874 (**5**), 2248864 [(-)-**10**], 2248861 [( + )-**14**], 2248865 [(-)-**14**], 2248863 (**15**), 2248858 [(-)-**17n**], 2248866 [(-)-**17o**]. These data can be obtained free of charge from the Cambridge Crystallographic Data Centre via www.ccdc.cam.ac.uk/data_request/cif. Whole Genome Sequencing (WGS) data have been deposited in the National Center for Biotechnology Information, under deposition numbers PRJNA1054306, PRJNA1052989, PRJNA1113534 and PRJNA1113431. These data can be obtained free of charge from the National Center for Biotechnology Information via https://www.ncbi.nlm.nih.gov/. The protein crystal structures (numbers PDB 7DUD and 5IS1) can be obtained free of charge from the https://www.rcsb.org/structure/7DUD and https://www.rcsb.org/structure/5IS1. Computational data are available within Supplementary Data 1. Chemical Experimental procedures, characterization of new compounds, and all other data supporting the findings are available in the manuscript and Supplementary Information. All data are available from the corresponding author upon request. Source data are provided with this paper.

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

## Acknowledgements

This work was supported by the National Natural Science Foundation of China [Nos. 82104019 to CM, 82373750 to WL, 82173859 to HW, 82293681(82293680) to YW], National Key R&D Program of China (No. 2022YFE0205500 to LC), the Guangdong Basic and Applied Basic Research Foundation (2024B1515040014 to WL), Key-Area Research and Development Program of Guangdong Province (2020B1111110004 to YW), Science and Technology Projects in Guangzhou (202102070001 to YW), and China Postdoctoral Science Foundation (2021M691260 to CM) and Shenzhen Science and Technology Major Projects (KJZD20230923115116032 to HW). We are grateful to Prof. Ren-Wang Jiang, and Dr. Wei Xu (Jinan Univ., Guagnzhou, China) for analyzing X-ray crystal structures. We are grateful to Prof. Yong-Heng, Wang and Mr. Xian-Bo, Chen (Jinan Univ., Guagnzhou, China) for analyzing theoretical calculations.

## Author contributions

M.J.C., Y.Y.W., T.H.Z., Y.X.H., and C.C.L. participated in chemical experiments and prepared the supplemental information. H.Z., S.Y.L, R.Q.C., C.X.H., Q.M.Z. and W.H. participated in biological experiments. L.W., W.C.Y., C.C.L. W.H., Q.M.Z., and M.J.C. conceived the project and wrote the manuscript. All the authors contributed to the discussion.

## Competing interests

The authors declare no competing interests.
