## [Peer Review File · Nature Communications]

Asymmetric Total Synthesis of Polycyclic Xanthenes and Discovery of a Walk Activator Active against MRSAEditorial Note: Parts of this Peer Review File have been redacted as indicated to remove third-party material where no permission to publish could be obtained.

REVIEWER COMMENTS

Reviewer #1 (Remarks to the Author):

This manuscript, authored by Wang and collaborators, presents the collective asymmetric total synthesis of polycyclic xanthenes, including myrtucommulone D and its related congeners. Empowered by the first successful synthesis of these natural products, this study has unveiled a novel antibacterial agent effective against both drug-sensitive and drug-resistant *S. aureus* strains. The stereoselective construction mechanism has been elucidated through density functional theory (DFT) calculations.

The author has devised a highly convergent strategy for synthesizing myrtucommulone D and its congeners, which involves chiral resolution from the racemic precursor. This approach enables the preparation of both R and S enantiomers with excellent enantiomeric excess. The author has also showcased the versatility of this method. Although the resolution strategy necessitates an additional two steps to obtain the chiral compound, the overall synthetic route proves to be robust.

The work presented in this manuscript carries significant implications from both a synthetic and practical standpoint, and the supplementary information is well-prepared. Given the importance of this research, I recommend its publication in Nature Communications once the suggested revisions have been addressed.

Regarding the main text:

1. Page 2: Please include relevant review references alongside Reference 10.
2. Page 2: For References 15, 16, 17, 18, and 20, clarify that these are not related to xanthenes and cite the correct papers. Additionally, consider citing the following papers: J.

Am. Chem. Soc. 2018, 140, 5065–5068. J. Am. Chem. Soc. 2011, 133, 9956–9959

3. The author's claim that hundreds of xanthenes have been identified in nature should be supported by relevant review references to guide the reader.

4. Page 3: Check the reference data of the MIC for Reference 22; it should be 1 µg/mL.

5. For Figure 1, provide highlights of this work alongside the first total synthesis.

6. Page 3: Review the style of References 26 and 27.

7. Page 4: Address the author's claim that the hemiketal moiety is critical for activity by adding relevant references.

8. Page 4: Verify the page number of Reference 32, it should read 13258-13263.

9. Page 6: Explain the differences between this work and the work referenced in 36. Also, provide possible reasons for the reaction not working.

10. Page 7: For the preparation of 15 and 16, explain why the yield improves through the crude product.

11. Define the chiral center in Figure 4.

12. Page 11: Include details about attempts to prepare compounds 1 through 19.

13. Page 11: The author proposed that a complex was formed between Cu(I) or Cu(II) and the carbonyl group at C8, leading to undesirable results. Provide evidence or relevant references to support this hypothesis.

14. Page 11: Indicate the corresponding SI Scheme to direct the reader.

15. Page 11: Clarify that the synthesis of 1 is not a single-step process but rather a two-step

reaction in one pot.

16. Page 12: Indicate the diastereomeric ratio (dr) of 19 in Figure 5a.

In regard to the Supplementary Information (SI), it is well-presented. However, the following aspects should be reviewed and addressed:

1. It would be beneficial to include NMR comparisons alongside the comparison table for synthetic and isolated compounds.

2. Please review the style of Reference 22.

3. Check the ¹³C NMR data, as it should read 101 MHz for a 400 MHz NMR instrument and 151 MHz for a 600 MHz NMR instrument.

4. Review the ¹³C NMR data for Compound S-2 (Page S5), which currently displays only 32 carbon signals compared to the 34 carbon atoms. If there are overlapping carbon signals, please indicate them. Additionally, verify the ¹³C NMR data for Compounds 16, 17ab, and 17a.

5. Check the optical rotation data of Compound S-4 and Compound S-5.

6. Define the chiral center in Table S1. The ChemDraw representation suggests a mixture; please clarify.

7. On Page S12, there is a missing dash in p-TsOH.

8. On Page S17, review the data for Compound 10b.

Reviewer #2 (Remarks to the Author):

Xanthenes, particularly those with polycyclic skeletons, have gained popularity in recent

total synthesis publications. As a nice addition to this body of literature, Wang, Huang, Ye, and Li et al. reported their completed asymmetric synthesis of the pentacyclic natural product myrtucommulone D and five related analogues with an unusual benzopyrano[2, 3-a]xanthene core. Of the five carbo- or heterocyclic rings within these molecules, the tricyclic xanthene moiety on the right of these molecules was constructed asymmetrically using an unusual Mitsunobu-mediated chiral resolution method. This approach exhibited a broad substrate scope and achieved excellent enantiomeric excess (92% to 99% ee). On the other hand, the left A/B bicyclic system was forged diastereoselectively via a successive retro-hemiketalization/double Michael cascade reaction. The interesting stereoselective transformation for constructing the bicyclic system was illustrated by Quantum mechanical calculations. Besides efficiently assembling the core skeleton, this work also demonstrated sophistication in the installation of the four stereocenters. The above accomplishment was by no means trivial as revealed by the failed attempts described in the Supplementary Information. The tactics and experiences gathered in the current synthetic work lay the foundation for the asymmetric synthesis of other complex polycyclic xanthenes.

Overall, the total synthesis described by the Wang/Huang/Ye/Li team is an impressive and inspiring achievement in the field of xanthene synthesis. In particular, there were 66 compounds in this work allowed the authors to conduct further studies on antibacterial activity. They discovered that compound 22 had a potent activity against MRSA *in vitro* and *in vivo*, comparable to that of vancomycin. Further genetic and biochemical studies suggested that this compound was a Walk activator, making it a promising antibacterial lead compound with a new mechanism. This is a nice achievement. Therefore, this work could be published in Nature Communications after correction of the following minor issues:

- (1) On page 11, it was mentioned that the L-proline was considered as a base to provide intermediate 19. Please explain whether it may also play a role of catalytic agent.
- (2) The result of antibacterial activity of *in vivo* is a nice achievement. Is it more appropriate to put the figure in the Supplementary Information (SI) into the body of literature?
- (3) In the SI, on page S4, the authors should explain why the retro-Friedel-Crafts of 10 led to the racemic S-1, but not the optical pure S-1.
- (4) On page 9, since the ee value was determined by chiral HPLC analysis, please add a note such as 'Determined by chiral HPLC analysis' in Figure 4.
- (5) On page 22, the general information of antibacterial activity should be added in

'Methods'.

(6) On page 23, 'et al' should not be italic, and the comma (,) after the volume number should not be bold in the 'References'.

Reviewer #3 (Remarks to the Author):

The paper of Cheng et al describes the total synthesis of testing of a range of myrtoCommulone D related compounds and their activity against both Gram-positive and Gram-negative bacteria. The authors make a case for their most active compound termed "22" is an activator of the conserved essential histidine kinase Walk. Below I highlight a number of issues that I have with the paper:

Figure 8a: Shows alignment of 18 amino acids surrounding the mutation identified in the erWalk for a spontaneously resistant mutant of *S. aureus* (SA22-SR). It is important to note that *Streptococcus pneumoniae* does not contain an extra cellular PAS domain and only has one transmembrane helix. There is no similarity of this region with *S. aureus* erWalk. Surprising as the MIC for *S. pneumoniae* for compound 22 is only double that of *S. aureus*, it would suggest that 22 is not specific for erWalk as is characterised in the paper and potentially has a second target

When compound 22 was tested, it was described as impacting MRSA, but all the subsequent characterisation was done in MSSA. Why?

Why was SPR not conducted with both the Walk Wt and R86C proteins? There is only in silico docking data for R86C / compound 22 affinity. SPR is established to look at this, why was it not used?. The R86C protein was purified for crystallisation.

Can the docking and SPR results be shown in the same units? Not obvious that the results are consistent (statement on page 20).

From the transcriptional data, SA22-SR is a down mutant, reduced transcription of all genes analysed which are positively regulated by WalkR. But due to the additional mutations present in the strain it is not possible to attribute the effect of the Walk R86C mutation on

the WalkR dependent regulation. Without this, specificity of 22 for Walk under biologically relevant conditions cannot be verified. The mutation needs to be recreated in the ATCC29213 Wt background to attribute the impact. Recently, Monk and Stinear (<https://doi.org/10.1099/acmi.0.000193>) published a method for allelic exchange with Walk used as an example. They have been successful introducing up and down mutations into Walk (<https://doi.org/10.1128/mbio.02262-23>, <https://doi.org/10.1038/s41467-019-10932-4>). Do other down mutants of Walk (eg. G223D) also have the same resistant phenotype. The Walk R86C mutation is in the literature (present with a second Walk mutation), so the process for allelic exchange should be successful (<https://doi.org/10.1038/srep17092>). It is surprising that there is no difference in the resistance to other antibiotics in the SA22-SR background as changes as resistance is well documented for recreated Walk or WalR down mutants.

Looking closer into the mutations present (and as the mutations present have been amalgamated, it is not possible to determine the co-occurrence) SA22-SR has a frame shift mutation in GlpK, this potentially would impact on the conversion glycerol to glycerol-3-phosphate. PlsY (indel in SA22-SR) uses glycerol-3-phosphate as the first step in lipid and lipoprotein biosynthesis. Is it possible that the resistance to compound 22 in SA22-SR is related to changes in membrane composition caused by these mutations?

Page 21: Stated in the discussion that these compounds are unlikely to yield resistance, but you obtain mutants that are resistant to the compound 22.

Why is CC48973 the MRSA strain used in the testing. No details on the strain eg clonal complex, antibiotic resistance profile – genome sequence. Later MRSA252 was used in in vivo assays. This genome sequenced strain should have been used in the testing or more details on CC48973 should be included. Same for CC49050 MSSA strain. Need more details. Same for the VRE and *S. pneumoniae* strains.

Table S11. Change Gram-negativestrains to Gram-negative strains. Your *Enterococcus faecalis* isolate is sensitive to vancomycin but the *Enterococcus faecium* vancomycin strain should be resistant – however upon testing it is also sensitive to vancomycin (needs to be

addressed). Would add in oxacillin results to show that the genotype of the MRSA and MSSA is correct. Would be good to have the ug/ml of each MIC shown along with uM.

Polymyxin B spelt incorrectly through-out.

In general the methods to not describe in enough detail to repeat the experiments. Some examples are shown below.

7.2 Antimicrobial agents and medium

What are the 5 antibacterial agents? Only 3 are mentioned.

7.3/7.4 Two different methods for the determination of the MIC. OD and MTT assay. Which was used?

7.5 – More details on the age/sex etc of the mice.

What was the vehicle?

What was the volume of the compound, vehicle or vancomycin applied to the micrfr?

How were the compound, vehicle or vancomycin applied?

3 mice per time point? Describe.

Why were MH agar plates used, and not TSA as previously described.

The way the serial dilutions are described would not dilute the cells enough to count the high numbers in day 1, 3 and 5 and 7. What was the limit of detection?

8.2 raw reads should be deposited rather than assemblies, to allow independent validation of the results.

What is the source of the ATCC29213 reference? Is it using the closed published genome. Or contigs from the illumina assembly.

More detail is need in the description of the method for the generation of spontaneous resistant mutants in the MSSA background. Why was ATCC29213 chosen for this when the emphasis has been on MRSA in the introduction? Four mutants were sequenced, need to highlight the mutations present in all these isolates. In the paper, why was that mutant chosen, do not mention the other 3. What mutations are present in each sequence isolate?

Table S14. Locus Tag has an asterisk but no description. Why was Newman used for the annotation?

8.3.1 Not enough detail. What primers were used? What was the method of cloning? What is the promoter driving expression?

Putting an essential histidine kinase on a plasmid (what is the plasmid copy number in your hands in SA22-SR?) can lead to unintended consequences through non-native levels of expression. The method that the gene was cloned into the pYJ335 was not described. It is an Anhydrotetracycline inducible vector. Was ATc used to induce expression? Does it complement other phenotypes? Have only shown the construct in a lysostaphin assay. Sheep blood hemolysis - hla is dramatically down. Does the addition of 22 to SA29213 increase SBA hemolysis? Increased alpha toxin expression in the presence of 22.

8.3.2 How was the RNA isolated? How was the data normalised? What method was used? What were the cells grown in? What growth stage was analysed? How long were the cells treated with lysostaphin for? What strains were compared?

How were the strains growth (temp, shaking speed?) What concentration of compound 22? Why is such a long exposure to compound 22 required? If it is activating Walk activity, would it not happen quickly? How much aeration?

The level of "biofilm" being formed in very low which is characteristic of some strains of *S. aureus*. What do the P values correspond to? related to the Wt+vehicle? Not explained.

No description of the, cloning, expression and minimal on the protein purification

Page 20, what reports are there of Walk activators?

REVIEWER COMMENTS

Reviewer #1 (Remarks to the Author):

This manuscript, authored by Wang and collaborators, presents the collective asymmetric total synthesis of polycyclic xanthenes, including myrtucommulone D and its related congeners. Empowered by the first successful synthesis of these natural products, this study has unveiled a novel antibacterial agent effective against both drug-sensitive and drug-resistant *S. aureus* strains. The stereoselective construction mechanism has been elucidated through density functional theory (DFT) calculations.

The author has devised a highly convergent strategy for synthesizing myrtucommulone D and its congeners, which involves chiral resolution from the racemic precursor. This approach enables the preparation of both R and S enantiomers with excellent enantiomeric excess. The author has also showcased the versatility of this method. Although the resolution strategy necessitates an additional two steps to obtain the chiral compound, the overall synthetic route proves to be robust.

The work presented in this manuscript carries significant implications from both a synthetic and practical standpoint, and the supplementary information is well-prepared. Given the importance of this research, I recommend its publication in Nature Communications once the suggested revisions have been addressed.

Regarding the main text:

1. Page 2: Please include relevant review references alongside Reference 10.
2. Page 2: For References 15, 16, 17, 18, and 20, clarify that these are not related to xanthenes and cite the correct papers. Additionally, consider citing the following papers: *J. Am. Chem. Soc.* 2018, 140, 5065 – 5068. *J. Am. Chem. Soc.* 2011, 133, 9956 – 9959
3. The author's claim that hundreds of xanthenes have been identified in nature should

be supported by relevant review references to guide the reader.

4. Page 3: Check the reference data of the MIC for Reference 22; it should be 1 $\mu\text{g}/\text{mL}$.

5. For Figure 1, provide highlights of this work alongside the first total synthesis.

6. Page 3: Review the style of References 26 and 27.

7. Page 4: Address the author's claim that the hemiketal moiety is critical for activity by adding relevant references.

8. Page 4: Verify the page number of Reference 32, it should read 13258-13263.

9. Page 6: Explain the differences between this work and the work referenced in 36. Also, provide possible reasons for the reaction not working.

10. Page 7: For the preparation of 15 and 16, explain why the yield improves through the crude product.

11. Define the chiral center in Figure 4.

12. Page 11: Include details about attempts to prepare compounds 1 through 19.

13. Page 11: The author proposed that a complex was formed between Cu(I) or Cu(II) and the carbonyl group at C8, leading to undesirable results. Provide evidence or relevant references to support this hypothesis.

14. Page 11: Indicate the corresponding SI Scheme to direct the reader.

15. Page 11: Clarify that the synthesis of 1 is not a single-step process but rather a two-step reaction in one pot.

16. Page 12: Indicate the diastereomeric ratio (dr) of 19 in Figure 5a.

In regard to the Supplementary Information (SI), it is well-presented. However, the following aspects should be reviewed and addressed:

1. It would be beneficial to include NMR comparisons alongside the comparison table for synthetic and isolated compounds.
2. Please review the style of Reference 22.
3. Check the ^{13}C NMR data, as it should read 101 MHz for a 400 MHz NMR instrument and 151 MHz for a 600 MHz NMR instrument.
4. Review the ^{13}C NMR data for Compound S-2 (Page S5), which currently displays only 32 carbon signals compared to the 34 carbon atoms. If there are overlapping carbon signals, please indicate them. Additionally, verify the ^{13}C NMR data for Compounds 16, 17ab, and 17a.
5. Check the optical rotation data of Compound S-4 and Compound S-5.
6. Define the chiral center in Table S1. The ChemDraw representation suggests a mixture; please clarify.
7. On Page S12, there is a missing dash in p-TsOH.
8. On Page S17, review the data for Compound 10b.

Reviewer #2 (Remarks to the Author):

Xanthenes, particularly those with polycyclic skeletons, have gained popularity in recent total synthesis publications. As a nice addition to this body of literature, Wang, Huang, Ye, and Li et al. reported their completed asymmetric synthesis of the pentacyclic natural product myrtucommulone D and five related analogues with an unusual benzopyrano[2, 3-a]xanthene core. Of the five carbo- or heterocyclic rings within these molecules, the tricyclic xanthene moiety on the right of these molecules was constructed asymmetrically using an unusual Mitsunobu-mediated chiral resolution method. This approach exhibited a broad substrate scope and achieved excellent enantiomeric excess (92% to 99% ee). On the other hand, the left A/B bicyclic system was forged diastereoselectively via a successive retro-hemiketalization/double Michael cascade reaction. The interesting stereoselective transformation for constructing the bicyclic system was illustrated by Quantum mechanical calculations. Besides efficiently assembling the core skeleton, this work also demonstrated sophistication in the installation of the four stereocenters. The above accomplishment was by no means trivial as revealed by the failed attempts described in the Supplementary Information. The tactics and experiences gathered in the current synthetic work lay the foundation for the asymmetric synthesis of other complex polycyclic xanthenes.

Overall, the total synthesis described by the Wang/Huang/Ye/Li team is an impressive and inspiring achievement in the field of xanthene synthesis. In particular, there were 66 compounds in this work allowed the authors to conduct further studies on antibacterial activity. They discovered that compound 22 had a potent activity against MRSA in vitro and in vivo, comparable to that of vancomycin. Further genetic and biochemical studies suggested that this compound was a WalK activator, making it a promising antibacterial lead compound with a new mechanism. This is a nice achievement. Therefore, this work could be published in Nature Communications after correction of the following minor issues:

- (1) On page 11, it was mentioned that the L-proline was considered as a base to provide intermediate 19. Please explain whether it may also play a role of catalytic agent.
- (2) The result of antibacterial activity of in vivo is a nice achievement. Is it more appropriate to put the figure in the Supplementary Information (SI) into the body of literature?
- (3) In the SI, on page S4, the authors should explain why the retro-Friedel-Crafts of 10 led to the racemic S-1, but not the optical pure S-1.
- (4) On page 9, since the ee value was determined by chiral HPLC analysis, please add a note such as 'Determined by chiral HPLC analysis' in Figure 4.
- (5) On page 22, the general information of antibacterial activity should be added in 'Methods'.
- (6) On page 23, 'et al' should not be italic, and the comma (,) after the volume number should not be bold in the 'References'.

Reviewer #3 (Remarks to the Author):

The paper of Cheng et al describes the total synthesis of testing of a range of myrtucommulone D related compounds and their activity against both Gram-positive and Gram-negative bacteria. The authors make a case for their most active compound termed "22" is an activator of the conserved essential histidine kinase Walk. Below I highlight a number of issues that I have with the paper:

Figure 8a: Shows alignment of 18 amino acids surrounding the mutation identified in the erWalk for a spontaneously resistant mutant of *S. aureus* (SA22-SR). It is important to note that *Streptococcus pneumoniae* does not contain an extra cellular PAS domain and only has one transmembrane helix. There is no similarity of this region with *S. aureus* erWalk. Surprising as the MIC for *S. pneumoniae* for compound 22 is only double that of *S. aureus*, it would suggest that 22 is not specific

for erWalK as is characterised in the paper and potentially has a second target

When compound 22 was tested, it was described as impacting MRSA, but all the subsequent characterisation was done in MSSA. Why?

Why was SPR not conducted with both the WalK Wt and R86C proteins? There is only in silico docking data for R86C / compound 22 affinity. SPR is established to look at this, why was it not used?. The R86C protein was purified for crystallisation.

Can the docking and SPR results be shown in the same units? Not obvious that the results are consistent (statement on page 20).

From the transcriptional data, SA22-SR is a down mutant, reduced transcription of all genes analysed which are positively regulated by WalR. But due to the additional mutations present in the strain it is not possible to attribute the effect of the WalK R86C mutation on the WalKR dependent regulation. Without this, specificity of 22 for WalK under biologically relevant conditions cannot be verified. The mutation needs to be recreated in the ATCC29213 Wt background to attribute the impact. Recently, Monk and Stinear (<https://doi.org/10.1099/acmi.0.000193>) published a method for allelic exchange with WalK used as an example. They have been successful introducing up and down mutations into WalK (<https://doi.org/10.1128/mbio.02262-23>, <https://doi.org/10.1038/s41467-019-10932-4>). Do other down mutants of WalK (eg. G223D) also have the same resistant phenotype. The WalK R86C mutation is in the literature (present with a second WalK mutation), so the process for allelic exchange should be successful (<https://doi.org/10.1038/srep17092>). It is surprising that there is no difference in the resistance to other antibiotics in the SA22-SR background as changes as resistance is well documented for recreated WalK or WalR down mutants.

Looking closer into the mutations present (and as the mutations present have been

amalgamated, it is not possible to determine the co-occurrence) SA22-SR has a frame shift mutation in GlpK, this potentially would impact on the conversion glycerol to glycerol-3-phosphate. PlsY (indel in SA22-SR) uses glycerol-3-phosphate as the first step in lipid and lipoprotein biosynthesis. Is it possible that the resistance to compound 22 in SA22-SR is related to changes in membrane composition caused by these mutations?

Page 21: Stated in the discussion that these compounds are unlikely to yield resistance, but you obtain mutants that are resistant to the compound 22.

Why is CC48973 the MRSA strain used in the testing. No details on the strain eg clonal complex, antibiotic resistance profile – genome sequence. Later MRSA252 was used in in vivo assays. This genome sequenced strain should have been used in the testing or more details on CC48973 should be included. Same for CC49050 MSSA strain. Need more details. Same for the VRE and *S. pneumoniae* strains.

Table S11. Change Gram-negative strains to Gram-negative strains. Your *Enterococcus faecalis* isolate is sensitive to vancomycin but the *Enterococcus faecium* vancomycin strain should be resistant – however upon testing it is also sensitive to vancomycin (needs to be addressed). Would add in oxacillin results to show that the genotype of the MRSA and MSSA is correct. Would be good to have the ug/ml of each MIC shown along with uM.

Polymyxin B spelt incorrectly through-out.

In general the methods to not describe in enough detail to repeat the experiments. Some examples are shown below.

7.2 Antimicrobial agents and medium

What are the 5 antibacterial agents? Only 3 are mentioned.

7.3/7.4 Two different methods for the determination of the MIC. OD and MTT assay.

Which was used?

7.5 – More details on the age/sex etc of the mice.

What was the vehicle?

What was the volume of the compound, vehicle or vancomycin applied to the micrfr?

How were the compound, vehicle or vancomycin applied?

3 mice per time point? Describe.

Why were MH agar plates used, and not TSA as previously described.

The way the serial dilutions are described would not dilute the cells enough to count the high numbers in day 1, 3 and 5 and 7. What was the limit of detection?

8.2 raw reads should be deposited rather than assemblies, to allow independent validation of the results.

What is the source of the ATCC29213 reference? Is it using the closed published genome. Or contigs from the illumina assembly.

More detail is need in the description of the method for the generation of spontaneous resistant mutants in the MSSA background. Why was ATCC29213 chosen for this when the emphasis has been on MRSA in the introduction? Four mutants were sequenced, need to highlight the mutations present in all these isolates. In the paper, why was that mutant chosen, do not mention the other 3. What mutations are present in each sequence isolate?

Table S14. Locus Tag has an asterisk but no description. Why was Newman used for the annotation?

8.3.1 Not enough detail. What primers were used? What was the method of cloning?

What is the promoter driving expression?

Putting an essential histidine kinase on a plasmid (what is the plasmid copy number in your hands in SA22-SR?) can lead to unintended consequences through non-native levels of expression. The method that the gene was cloned into the pYJ335 was not described. It is an Anhydrotetracycline inducible vector. Was ATc used to induce expression? Does it complement other phenotypes? Have only shown the construct in a lysostaphin assay. Sheep blood hemolysis - hla is dramatically down. Does the addition of 22 to SA29213 increase SBA hemolysis? Increased alpha toxin expression in the presence of 22.

8.3.2 How was the RNA isolated? How was the data normalised? What method was used? What were the cells grown in? What growth stage was analysed? How long were the cells treated with lysostaphin for? What strains were compared?

How were the strains growth (temp, shaking speed?) What concentration of compound 22? Why is such a long exposure to compound 22 required? If it is activating Walk activity, would it not happen quickly? How much aeration?

The level of "biofilm" being formed is very low which is characteristic of some strains of *S. aureus*. What do the P values correspond to? related to the Wt+vehicle? Not explained.

No description of the, cloning, expression and minimal on the protein purification

Page 20, what reports are there of Walk activators?

Response Letter for Manuscript “Asymmetric Total Synthesis of Polycyclic Xanthenes and Discovery of the First Walk Activator with Potent Activity against MRSA (NCOMMS-23-36758)”

We highly appreciate the referees for their constructive and detailed reviews. We have supplemented relevant experiments and revised the manuscript in accordance with all their comments. The positive changes in both the manuscript text file and supplementary information have been marked with track changes or “yellow colour highlighting” facility. We have so indicated in our point-by-point response and revision summary below.

I Response to Reviewer 1:

Comments: This manuscript, authored by Wang and collaborators, presents the collective asymmetric total synthesis of polycyclic xanthenes, including myrtucommulone D and its related congeners. Empowered by the first successful synthesis of these natural products, this study has unveiled a novel antibacterial agent effective against both drug-sensitive and drug-resistant *S. aureus* strains. The stereoselective construction mechanism has been elucidated through density functional theory (DFT) calculations.

The author has devised a highly convergent strategy for synthesizing myrtucommulone D and its congeners, which involves chiral resolution from the racemic precursor. This approach enables the preparation of both R and S enantiomers with excellent enantiomeric excess. The author has also showcased the versatility of this method. Although the resolution strategy necessitates an additional two steps to obtain the chiral compound, the overall synthetic route proves to be robust.

The work presented in this manuscript carries significant implications from both a synthetic and practical standpoint, and the supplementary information is well-prepared. Given the importance of this research, I recommend its publication in Nature Communications once the suggested revisions have been addressed.

Answer: We highly appreciate the reviewer's comments and suggestions on our manuscript.

Question 1: Page 2: Please include relevant review references alongside Reference 10.

Answer 1: Thank you for your suggestion. We have cited the following three review references related to “particularly xanthenes with antibacterial activity” (please see page 2, references 11-13 in the revised manuscript).

References:

11. Miladiyah, I. & Rachmawaty, F. J. Potency of xanthone derivatives as antibacterial agent against methicillin-resistant *Staphylococcus aureus* (MRSA). *JKKI: Jurnal Kedokteran dan Kesehatan Indonesia* **8**, 124-135 (2017).
12. Araújo, J., Fernandes, C., Pinto, M. & Tiritan, M. E. Chiral derivatives of xanthenes with antimicrobial activity. *Molecules* **24**, 314 (2019).
13. Liu, X., Shen, J. & Zhu, K. Antibacterial activities of plant-derived xanthenes. *RSC Med. Chem.* **13**, 107-116 (2022).

Question 2: Page 2: For References 15, 16, 17, 18, and 20, clarify that these are not related to xanthenes and cite the correct papers. Additionally, consider citing the following papers: *J. Am. Chem. Soc.* 2018, 140, 5065–5068. *J. Am. Chem. Soc.* 2011, 133, 9956–9959

Answer 2: Thank you for pointing out this problem. We have checked out the references in the revised manuscript, and cited the following references (please see page 2, references 14-21 in the revised manuscript).

This two works regarding on total synthesis of xanthenes [*J. Am. Chem. Soc.* **140**, 5065–5068 (2018).; *J. Am. Chem. Soc.* **133**, 9956–9959 (2011).] are very important. We have cited the two references in the revised manuscript (please see references 15 and 19).

References:

14. Müller, H. et al. Total synthesis of myrtucommulone A. *Angew. Chem. Int. Ed.* **49**, 2045-2049 (2010).
15. Butler, J. R., Wang, C., Bian, J. & Ready, J. M. Enantioselective total synthesis of (-)-kibdelone C. *J. Am. Chem. Soc.* **133**, 9956-9959 (2011).
16. Axelrod, A., Eliassen, A. M., Chin, M. R., Zlotkowski, K. & Siegel, D. Syntheses of xanthofulvin and vinaxanthone, natural products enabling spinal cord regeneration. *Angew. Chem. Int. Ed.* **52**, 3421-3424 (2013).
17. Qin, T. et al. Atropselective syntheses of (-)- and (+)-rugulotrosin A utilizing point-to-axial chirality transfer. *Nat. Chem.* **7**, 234-240 (2015).
18. Yang, J. et al. Approaches to polycyclic 1, 4-dioxygenated xanthenes. Application to total synthesis of the aglycone of IB-00208. *Org. Lett.* **17**, 114-117 (2015).
19. Holmbo, S. D. & Pronin, S. V. A concise approach to anthraquinone-xanthone heterodimers. *J. Am. Chem. Soc.* **140**, 5065-5068 (2018).
20. Ito, S. et al. Total synthesis of termicalcicolanone A via organocatalysis and regioselective Claisen rearrangement. *Org. Lett.* **21**, 2777-2781 (2019).
21. Xie, T., Zheng, C., Chen, K., He, H. B. & Gao, S. H. Asymmetric total synthesis of the complex polycyclic xanthone FD-594. *Angew. Chem. Int. Ed.* **132**, 4390-4394 (2020).

Question 3: The author's claim that hundreds of xanthenes have been identified in nature should be supported by relevant review references to guide the reader.

Answer 3: Thank you for your suggestion. We have cited the relevant four review references as follows (please see page 2, references 22-25 in the revised manuscript).

22. Masters, K. S. & Bräse, S. Xanthenes from fungi, lichens, and bacteria: the natural products and their synthesis. *Chem. Rev.* **112**, 3717-3776 (2012).
23. Winter, D. K., Sloman, D. L. & Porco Jr, J. A. Polycyclic xanthone natural products: structure, biological activity and chemical synthesis. *Nat. Prod. Rep.* **30**, 382-391 (2013).

24. Nicoletti, R. et al. Structures and bioactive properties of myrtucommulones and related acylphloroglucinols from Myrtaceae. *Molecules* **23**, 3370 (2018).

25. Kong, L., Deng, Z. & You, D. Chemistry and biosynthesis of bacterial polycyclic xanthone natural products. *Nat. Prod. Rep.* **39**, 2057-2095 (2022).

Question 4: Page 3: Check the reference data of the MIC for Reference 22; it should be 1 $\mu\text{g}/\text{mL}$.

Answer 4: Thank you for your suggestion. We have checked the reference data [*Chem. Biodivers.* **17**, e2000292 (2020)]. The MIC of compound **1** is 2 $\mu\text{g}/\text{mL}$, as described in the manuscript.

Question 5: For Figure 1, provide highlights of this work alongside the first total synthesis.

Answer 5: Thank you for your helpful suggestion. We have added “Novel *Retro*-hemiketalization/double Michael cascade reaction” and “Unique Mitsunobu-mediated chiral resolution” into the highlights in the revised manuscript (please see the following **Figure I**).

Figure I. Structure of **1** and the highlights

Question 6: Page 3: Review the style of References 26 and 27.

Answer 6: Thank you for your help. We have checked and revised the style of these references (please see references 31 and 32 in the revised manuscript).

Question 7: Page 4: Address the author's claim that the hemiketal moiety is critical for activity by adding relevant references.

Answer 7: Thank you for your suggestion. We have added the relevant references (please see page 4, references 27 and 33) into the revised manuscript. According to the references, the natural products with hemiketal moiety (**Figure II, A**) displayed better antibacterial activity than the dehydration products (**Figure II, B**), indicating the hemiketal moiety (highlighted in yellow) was critical for antibacterial activity.

Figure redacted

Figure II. Structures of A and B

Question 8: Page 4: Verify the page number of Reference 32, it should read 13258-13263.

Answer 8: Thank you for your help. We have revised the page number of this reference (please see reference 37 in the revised manuscript).

Question 9: Page 6: Explain the differences between this work and the work referenced in 36. Also, provide possible reasons for the reaction not working.

Answer 9: Thank you for your suggestion. We have added these explanations into the revised SI.

1. The differences between this work and the work in Ref. 36 [*Chem. Sci.* **9**, 1488-1495 (2018)].

(1) In this work, asymmetric synthesis of 2- and 4-substituted xanthenes (**Figure IIIa**) can be readily realized within 12 h from the *rac*-xanthenes and inexpensive chiral alcohol (~40 \$/50 g) by a Mitsunobu-mediated chiral resolution. However, in our

previous work, the asymmetric Friedel-Crafts-type Michael addition could only give 2-substituted xanthenes, and need an expensive chiral phosphoric acid [(S)-C; ~400 \$/100 mg] with long time (up to 7 days) (**Figure IIIb**, the work referenced in [*Chem. Sci.* **9**, 1488-1495 (2018)]).

(2) In this work, the chiral products (4- and 2-substituted xanthenes) were not only translated into the natural products **1** and its analogues (**Figure IIIa**) but also served as precursors for the asymmetric synthesis of myrtucommuacetalone B (**Figure IIIb**) and other related natural products. However, the work referenced in “*Chem. Sci.* **9**, 1488-1495 (2018)” could hardly achieve the asymmetric synthesis of **1** and its analogues.

(3) In this work, the three contiguous stereocenters in **1** were constructed using a novel *retro*-hemiketalization-double Michael addition cascade reaction (**Figure IIIa**, *dr* > 20:1), which provided a single product **1** in 81% yield. The diastereoselective reaction mechanism was elucidated through experiments and Quantum mechanical calculations. However, constructing the consecutive three stereocenters in myrtucommuacetalone B, as referenced in “*Chem. Sci.* **9**, 1488-1495 (2018)”, using a Michael-ketalization-annulation cascade reaction resulted in poorer diastereoselective (*dr* 11:1) compared to the result in this work. Meanwhile, the reaction mechanism has not yet been explained through experiments or Quantum mechanical calculations (**Figure IIIb**).

Figure redacted

Figure III. a, This work. b, The work referenced in 36 [*Chem. Sci.* 9, 1488-1495 (2018)].

2. The possible reasons for the reaction not working.

(1) Although the 2-substituted xanthone **10** obtained from the reference [*Chem. Sci.* 9, 1488-1495 (2018)] can undergo a Michael addition reaction to yield **22**, the transformation of **22** to **5** [*Angew. Chem. Int. Ed.* 49, 2045-2049 (2010).] presents a challenge (**Figure IVa**). This is due to the likelihood of **22** existing in a linear form

under acidic conditions, leading to the intramolecular attack of the less hindered free phenolic hydroxyl group (C4-OH) on the less hindered C6'' and resulting in the formation of the linear product S-5.

(2) An optically pure 2-substituted xanthene, exemplified by compound **11** (**Figure IVb**) and obtainable according to the procedures outlined in the reference [*Chem. Sci.* **9**, 1488-1495 (2018)], proved challenging to convert into optically pure compound **13** through *retro*-Friedel-Crafts, followed by Friedel-Crafts reactions. This hindrance was attributed to the issue of racemization [*J. Org. Chem.* **87**, 4788-4800 (2022)] (**Figure IVb**), making it difficult to achieve the asymmetric synthesis of compound **1** and its analogues. Additionally, the proposed racemization mechanism has been discussed in the revised SI (please see SI Scheme S3c).

(3) The novel *retro*-hemiketalization-double Michael addition cascade reaction, crucial for the synthesis of compound **1** and its analogues, is absent in the reference [*Chem. Sci.* **9**, 1488-1495 (2018)]. Consequently, the absence of this key reaction in this reference may be a contributing factor to the challenge of achieving compound **1** and its analogues.

Figure redacted

Figure IV. Reported attempts to synthesis of myrtucommulone E and optically pure rhodomyrtone.

Question 10: Page 7: For the preparation of **15** and **16**, explain why the yield improves through the crude product.

Answer 10: Thank you for your suggestion. We improve the yield not through the crude products. The yield could be improved by replacing the solvent PhMe with THF/PhMe (v/v = 1:1). We have included detailed information on this modification in the revised manuscript (please see page 7). The use of this mixed solvent has been shown to significantly reduce the formation of Friedel-Crafts-type byproducts (**Figure V**).

Figure V. The structures of the byproducts.

Question 11: Define the chiral center in Figure 4.

Answer 11: Thank you for your suggestion. We have defined the chiral centers in Figure 4 in the revised manuscript.

Original version

Revised version

Question 12: Page 11: Include details about attempts to prepare compound **1** through **19**.

Answer 12: Thank you for your suggestion. In the revised manuscript, we have included details about our attempts to prepare compound **1** through **19** (please see pages 11-12).

Original version

Accordingly, after extensive investigation, gratifyingly, treatment of **19** with *i*PrMgBr (3.5 equiv.) and an excess amount of CuCN (1.1 equiv.) in THF/CH₂Cl₂ afforded the (+)-myrtucommulone D (**1**) in 81% yield without the need for protecting groups.

Revised version

Gratifyingly, treatment of **19** with *i*PrMgBr (3.5 equiv.) and an excess of CuI (1.1 equiv.) in THF at -50 °C afforded the desired (+)-myrtucommulone D (**1**) in 18% yield. Meanwhile, some starting materials were recovered. Encouraged by this result, we further explored a variety of solvents including CH₂Cl₂, PhMe, Et₂O, 1,4-dioxane and THF/CH₂Cl₂. Rewardingly, using THF/CH₂Cl₂ instead of THF as the solvent accelerated the transformation, increasing the yield of (+)-**1** to 58%. Subsequent screening of copper reagents (CuI, CuBr, CuCN, CuBr·SMe₂) revealed that CuCN provided the most excellent yield for **1**. After extensive investigation, the optimal protocol was identified: when **19** was treated with *i*PrMgBr (3.5 equiv.) in the

presence of CuCN (1.1 equiv.) in THF/CH₂Cl₂ at -78 °C to -50 °C, (+)-**1** was obtained in 81% yield without the need for protecting groups.

Question 13: Page 11: The author proposed that a complex was formed between Cu(I) or Cu(II) and the carbonyl group at C8, leading to undesirable results. Provide evidence or relevant references to support this hypothesis.

Answer 13: Thank you for your suggestion. We have cited the following two relevant references that describe the formation of complexes involving Cu(I) or Cu(II) with the carbonyl group and an oxyanion, respectively, to support our hypothesis (references 63 and 64 in the revised manuscript).

References:

63. Lang, H. et al. Mono- and bimetallic copper (I)- and silver (I)-phosphane complexes with β -diketonate units. *Z. Anorg. Allg. Chem.* **629**, 2371-2380 (2003).

64. Yang, A. et al. A multifunctional anti-AD approach: design, synthesis, X-ray crystal structure, biological evaluation and molecular docking of chrysin derivatives. *Eur. J. Med. Chem.* **233**, 114216 (2022).

Question 14: Page 11: Indicate the corresponding SI Scheme to direct the reader.

Answer 14: Thank you for your suggestion. We have added the all corresponding SI schemes, tables, Figures or pages in the revised manuscript.

Question 15: Page 11: Clarify that the synthesis of **1** is not a single-step process but rather a two-step reaction in one pot.

Answer 15: Thank you for pointing out this problem. We have revised “in a single step by this new cascade reaction” to “by a two-step reaction in one pot” (please see page 13 in the revised manuscript).

Question 16: Page 12: Indicate the diastereomeric ratio (*dr*) of **19** in Figure 5a.

Answer 16: Thank you for your suggestion. We have added the diastereomeric ratio (*dr*) of **19** in Figure 5a (please see page 12 in the revised manuscript).

Comments: In regard to the Supplementary Information (SI), it is well-presented. However, the following aspects should be reviewed and addressed.

Question 17: It would be beneficial to include NMR comparisons alongside the comparison table for synthetic and isolated compounds.

Answer 17: Thank you for your suggestion. In the Supplementary Information (SI), alongside the comparison table, we have added NMR comparison pictures on images S222, S227, S231, S235, and S237.

Question 18: Please review the style of Reference 22.

Answer 18: We have revised the style of reference 22 in the SI.

Original version

Laskowski, R. A., MacArthur, M. W., Moss, D. S. & Thornton, J. M. PROCHECK: a program to check the stereochemical quality of protein structures. *Journal of Applied Crystallogr.* **26**, 283-291 (1993).

Revised version

Laskowski, R. A., MacArthur, M. W., Moss, D. S. & Thornton, J. M. PROCHECK: a program to check the stereochemical quality of protein structures. *J. Appl. Crystallogr.* **26**, 283-291 (1993).

Question 19: Check the ^{13}C NMR data, as it should read 101 MHz for a 400 MHz NMR instrument and 151 MHz for a 600 MHz NMR instrument.

Answer 19: Thank you for your suggestion. We have checked the ^{13}C NMR data and adjusted the frequencies of 100 MHz, 125 MHz and 150 MHz to 101 MHz, 126 MHz and 151 in the revised SI, respectively.

Question 20: Review the ^{13}C NMR data for Compound **S-2** (Page S5), which currently displays only 32 carbon signals compared to the 34 carbon atoms. If there are overlapping carbon signals, please indicate them. Additionally, verify the ^{13}C NMR data for Compounds **16**, **17ab**, and **17a**.

Answer 20: Thank you for your help. We have reviewed the ^{13}C NMR data for all compounds. As you noted, there are overlapping carbon signals in compounds, such as *rac*-**S-2**, **16**, **17ab**, **17a**, and others. We have currently identified and documented these overlapping carbon signals in the SI on pages S6, S8-S9, S20-S21, S24-S25, S27-S29, S31-S32, S34-S36, S38-S39, S41-S42, S44-S45, S47-S48, S54-S55, S50-S52, S57-S58, S60-S61, S64, S67-S68, S70-S71, S74-S77, S79-S80, S84-S85, S97, S100.

Question 21: Check the optical rotation data of compound **S-4** and compound **S-5**.

Answer 21: Thank you for pointing out the issue in the SI. We have now revised the optical rotation data for compounds **S-4** and **S-5** on pages S8 and S9 in the updated SI.

Original version

Compound **S-4**: $[\alpha]_{\text{D}}^{25} = +81.6$ ($c = 0.1$ in MeOH)

Compound **S-5**: $[\alpha]_{\text{D}}^{25} = 0$ ($c = 0.1$ in MeOH)

Revised version

Compound **S-4**: $[\alpha]_{\text{D}}^{25} = 0$ ($c = 0.1$ in MeOH)

Compound **S-5**: $[\alpha]_{\text{D}}^{25} = +81.6$ ($c = 0.1$ in MeOH)

Question 22: Define the chiral center in Table S1. The ChemDraw representation suggests a mixture; please clarify.

Answer 22: Thank you for your suggestion. We have defined the chiral center in Table S1 (please see page S14).

Question 23: On Page S12, there is a missing dash in p-TsOH.

Answer 23: Thank you for pointing out this problem in the SI. On page S15, we have revised the *p*TsOH to *p*-TsOH.

Question 24: On Page S17, review the data for Compound **10b**.

Answer 24: Thank you for your suggestion. We have checked the data for compound **10b** and identified overlapping carbon signals. We have now documented the overlapping carbon signals for all compounds in the SI. Additionally, on page S21 of the revised SI, we have corrected “Compound **10**” and “a total of 2.3 g of compound **10a**” to “Compound **10b**” and “a total of 2.3 g of compound **10b**”, respectively.

II Response to Reviewer 2:

Comments: Xanthenes, particularly those with polycyclic skeletons, have gained popularity in recent total synthesis publications. As a nice addition to this body of literature, Wang, Huang, Ye, and Li et al. reported their completed asymmetric synthesis of the pentacyclic natural product myrtucommulone D and five related analogues with an unusual benzopyrano[2, 3-*a*]xanthene core. Of the five carbo- or heterocyclic rings within these molecules, the tricyclic xanthene moiety on the right of these molecules was constructed asymmetrically using an unusual Mitsunobu-mediated chiral resolution method. This approach exhibited a broad substrate scope and achieved excellent enantiomeric excess (92% to 99% ee). On the other hand, the left A/B bicyclic system was forged diastereoselectively via a successive retro-hemiketalization/double Michael cascade reaction. The interesting stereoselective transformation for constructing the bicyclic system was illustrated by Quantum mechanical calculations. Besides efficiently assembling the core skeleton, this work also demonstrated sophistication in the installation of the four stereocenters. The above accomplishment was by no means trivial as revealed by the failed attempts described in the Supplementary Information. The tactics and experiences gathered in

the current synthetic work lay the foundation for the asymmetric synthesis of other complex polycyclic xanthenes.

Overall, the total synthesis described by the Wang/Huang/Ye/Li team is an impressive and inspiring achievement in the field of xanthene synthesis. In particular, there were 66 compounds in this work allowed the authors to conduct further studies on antibacterial activity. They discovered that compound **22** had a potent activity against MRSA in vitro and in vivo, comparable to that of vancomycin. Further genetic and biochemical studies suggested that this compound was a WalK activator, making it a promising antibacterial lead compound with a new mechanism. This is a nice achievement. Therefore, this work could be published in *Nature Communications* after correction of the following minor issues.

Answer: We highly appreciate the reviewer's comments and suggestions on our manuscript.

Question 1: On page 11, it was mentioned that the L-proline was considered as a base to provide intermediate **19**. Please explain whether it may also play a role of catalytic agent.

Answer 1: Thank you for your suggestion. We believe that L-proline should play a role of base. The reason is as follows: when **18** was treated with **8** using a catalytic amount (0.2 equiv.) of L-proline, the yield of **19** was less than 10% in 48 h, and most of **18** was recovered (**Figure V**). However, as the amount of L-proline increased, the yield also improved. For example, using 1.0 equiv. of L-proline resulted in a yield of **19** at 86%. Taken together, L-proline is considered a base that enhances the nucleophilicity of **8**, thereby promoting the reaction.

Figure V. Synthesis of **19**.

Question 2: The result of antibacterial activity of *in vivo* is a nice achievement. Is it more appropriate to put the figure in the Supplementary Information (SI) into the body of manuscript?

Answer 2: Thank you for your suggestion. As the results of the animal experiments have been extensively described in the main text, in order to save space, we recommend placing the relevant images in the Supporting Information.

Question 3: In the SI, on page S4, the authors should explain why the *retro*-Friedel-Crafts of **10** led to the racemic **S-1**, but not the optical pure **S-1**.

Answer3: Thank you for your suggestion. The racemization in this reaction may be attributed to the formation of intermediate II, which undergoes a 1,3-hydrogen shift to yield III, resulting in racemization (**Figure VI**). The detailed reaction process is provided below, and it has been included in the revised SI (please see page S4).

Figure VI. The proposed racemization mechanism of compound **10** during the

retro-Friedel-Crafts reaction.

Question 4: On page 9, since the *ee* value was determined by chiral HPLC analysis, please add a note such as ‘Determined by chiral HPLC analysis’ in Figure 4.

Answer 4: Thank you for your suggestion. We have added “The *ee* values were determined by chiral HPLC analysis” in Figure 4 and Table S1, respectively (please see pages 10 in the revised manuscript, and S15 in the revised SI).

Question 5: On page 22, the general information of antibacterial activity should be added in ‘Methods’.

Answer 5: Thank you for your suggestion. Considering that the general information of “antibacterial activity” will occupy a substantial portion in the manuscript, we have added “the general information of antibacterial activity” in the revised SI (please see pages S240-265).

Question 6: On page 23, ‘*et al*’ should not be italic, and the comma (,) after the volume number should not be bold in the ‘References’.

Answer 6: Thank you for pointing out these problems in the manuscript. We have revised italic “*et al*” and bold comma (,) to be non-italic “et al” and non-bold comma (,), respectively. We have also revised other errors in the “References” (please see pages 24-30 in the revised manuscript)

III Response to Reviewer 3:

Comments: The paper of Cheng et al describes the total synthesis of testing of a range of myrtucommulone D related compounds and their activity against both Gram-positive and Gram-negative bacteria. The authors make a case for their most active compound termed “**22**” is an activator of the conserved essential histidine kinase WalK. Below I highlight a number of issues that I have with the paper:

Answer: We highly appreciate the reviewer’s comments and suggestions on our manuscript.

Question 1: Figure 8a: Shows alignment of **18** amino acids surrounding the mutation identified in the erWalK for a spontaneously resistant mutant of *S. aureus* (SA_{22-SR}). It is important to note that *Streptococcus pneumoniae* does not contain an extra cellular PAS domain and only has one transmembrane helix. There is no similarity of this region with *S. aureus* erWalK. Surprising as the MIC for *S. pneumoniae* for compound **22** is only double that of *S. aureus*, it would suggest that **22** is not specific for erWalK as is characterised in the paper and potentially has a second target.

Answer 1: Thank you for your suggestion. We focused solely on sequence similarity and were not aware of the transmembrane structural features of the *Streptococcus pneumoniae* WalK protein. Due to the lack of similarity to *S. aureus* erWalK, we realized that it was no longer appropriate to include the results of the sequence alignment in the manuscript. Consequently, we have removed this section in the revised manuscript (please see Figure 8).

We strongly agree with your speculation that the inhibitory effect of **22** on *Streptococcus pneumoniae* suggests that it may have another target. This is further supported by the difference in resistance levels observed between SA_{WalK(R86C)} (SA₂₉₂₁₃ carrying only R86C point mutation) and SA_{22-SR} strains to **22**. Considering the mutation sites in essential genes, such as *plsY* in SA_{22-SR} strains, it further supports that compound **22** should have more than one target. As for other targets of **22**, we will conduct more in-depth research in the follow-up study. The revised manuscript now includes a description of the aforementioned content (please see lines 468 to 476 in the manuscript).

Question 2: When compound **22** was tested, it was described as impacting MRSA, but all the subsequent characterisation was done in MSSA. Why?

Answer 2: Thank you for your suggestion. The significance of **22** lies primarily in its efficacy against MRSA, prompting us to assess its antibacterial activity *in vitro* and *in vivo* using MRSA strains. In studies of the mode of action, we typically utilize standard strains with a clear genetic background. For example, in a related study on

the mode of action of silver against MRSA (*Nat Commun.* **12**, 3331 (2021). <https://doi.org/10.1038/s41467-021-23659-y>), the Newman strain (MSSA) was employed. The use of MSSA strains did not compromise our understanding of the mechanism of action or the reasons behind the compound's activity against MRSA. For example, **22** can activate the function of WalK, and MRSA does not exhibit resistance to this mode of action, potentially explaining the effectiveness of **22** against MRSA.

Question 3: Why was SPR not conducted with both the WalK Wt and R86C proteins? There is only in silico docking data for R86C/compound **22** affinity. SPR is established to look at this, why was it not used? The R86C protein was purified for crystallisation.

Answer 3: Thank you for your suggestion. We strongly agree with you. In the revised manuscript, we added the SPR results for the R86C protein with **22**. These results reveal a significant decrease in the affinity of erWalK_{R86C} for **22** when compared to erWalK (please see SI pages S261-S262, and Figure S10).

Question 4: Can the docking and SPR results be shown in the same units? Not obvious that the results are consistent (statement on page 20).

Answer 4: Thank you for your suggestion. We adjusted the units of docking to be consistent with SPR (please see lines 446 to 449 in the manuscript, and Figure S10 in the revised SI).

Question 5: From the transcriptional data, SA_{22-SR} is a down mutant, reduced transcription of all genes analysed which are positively regulated by WalR. But due to the additional mutations present in the strain it is not possible to attribute the effect of the WalK R86C mutation on the WalKR dependent regulation. Without this, specificity of **22** for WalK under biologically relevant conditions cannot be verified. The mutation needs to be recreated in the ATCC29213 Wt background to attribute the impact. Recently, Monk and Stinear (<https://doi.org/10.1099/acmi.0.000193>)

published a method for allelic exchange with WalK used as an example. They have been successful introducing up and down mutations into WalK (<https://doi.org/10.1128/mbio.02262-23>, <https://doi.org/10.1038/s41467-019-10932-4>). Do other down mutants of WalK (eg. G223D) also have the same resistant phenotype. The WalK R86C mutation is in the literature (present with a second WalK mutation), so the process for allelic exchange should be successful (<https://doi.org/10.1038/srep17092>). It is surprising that there is no difference in the resistance to other antibiotics in the SA_{22-SR} background as changes as resistance is well documented for recreated WalK or WalR down mutants.

Answer 5: Thank you for your suggestion. In the revised manuscript, we performed CRISPR-Cas9 and successfully recreated the R86C mutation in the ATCC29213 background (SA_{walK(R86C)}). The results revealed decreased transcription levels of *lytM*, *hla*, *ssaA*, *sdrD*, *ebpS*, *sceD*, *SA0710*, *SA2097* and *SA2353* in SA_{walK(R86C)}, indicating down-regulation of WalK function. However, the R86C mutation in the *walK* gene is highly unstable and usually spontaneously reverses to the wild state after 2-3 passages of culture. This may explain the coexistence of various other mutations in resistant strains and the absence of the R86C mutation alone in clinical isolates.

We have found from previous studies that mutations at different positions of WalK have different effects on antibiotic susceptibility. In a literature reference (<https://doi.org/10.1038/srep17092>), the MIC of vancomycin against VR4 (R86C and I287T), VR8 (A582E) and VR-RN (M426I) were 32, 16 and 32 µg/ml, respectively. In another literature (<https://doi.org/10.1016/j.ijantimicag.2019.08.021>), the MIC of vancomycin against *S. aureus* with L7Q mutation of WalK was 4 µg/ml. In the current investigation, we found that the MIC of vancomycin in SA_{walK(R86C)} was 2 µg/ml, suggesting that other point mutations in SA_{22-SR} strains also had an effect on vancomycin susceptibility. We have discussed the details in the revised manuscript (please see lines 412 to 423).

Question 6: Looking closer into the mutations present (and as the mutations present have been amalgamated, it is not possible to determine the co-occurrence) SA_{22-SR} has

a frame shift mutation in GlpK, this potentially would impact on the conversion glycerol to glycerol-3-phosphate. PlsY (indel in SA_{22-SR}) uses glycerol-3-phosphate as the first step in lipid and lipoprotein biosynthesis. Is it possible that the resistance to compound **22** in SA_{22-SR} is related to changes in membrane composition caused by these mutations?

Answer 6: Thank you for your suggestion. We strongly agree with your inference regarding the relationship between the changes in membrane composition caused by Glpk and PlsY mutations and the resistance to compound **22**. Among the mutations of SA_{22-SR}, *walk* and *plsY* are essential for the survival, so it is reasonable to believe that these two genes are responsible for the inhibitory activity of **22**. In the present study, we specifically focused on the effects of **22** on WalK function. Exploring the role played by mutations in *plsY* and its functionally related *glpK* in resistance to **22**, as well as the impact of **22** on PlsY function, is a promising avenue for future research and could be pursued as a separate project. Additionally, considering that natural products always have multiple targets [*Eur. J. Med. Chem.* **163**, 911-931 (2019); *Bioorgan. Med. Chem.* **43**, 116270 (2021); *Sci. Adv.* **9**, eadg5995 (2023) etc.], more studies should be conducted to fully elucidate the antimicrobial mechanism of **22**. We have incorporated this discussion into the revised manuscript (please see lines 468 to 476).

Question 7: Page 21: Stated in the discussion that these compounds are unlikely to yield resistance, but you obtain mutants that are resistant to the compound **22**.

Answer 7: Thank you for your suggestion. Induction of spontaneous drug resistance is a common technique in the study of antimicrobial mechanisms. Many articles have employed this approach to identify targets for antimicrobial agents (<https://doi.org/10.1038/s41586-023-06873-0>. *Nature*. 2024 Jan 3. Online ahead of print). The availability of resistant strains does not imply easy resistance. In the revised SI, we compared the time of emergence of resistance in *S. aureus* induced by **22** and Norfloxacin. The results showed that the time of induction of drug resistance by **22** was significantly longer than that by Norfloxacin (please see SI page S245, and

Figure S2).

Question 8: Why is CC48973 the MRSA strain used in the testing. No details on the strain eg clonal complex, antibiotic resistance profile – genome sequence. Later MRSA252 was used in *in vivo* assays. This genome sequenced strain should have been used in the testing or more details on CC48973 should be included. Same for CC49050 MSSA strain. Need more details. Same for the VRE and *S. pneumoniae* strains.

Answer 8: Thank you for your suggestion. We opted for clinical strains to showcase the potential clinical application of compound **22**. Acknowledging your point, we realize that we neglected to provide background information on the clinical strains used. In response, we have uploaded the genome sequences of all clinical strains to NCBI and provided the BioProject number in the revised manuscript (please see supplementary method section 7.1). Additionally, we have incorporated the testing of MRSA 252 in the revised manuscript (please see SI Table S11).

Question 9: Table S11. Change Gram-negative strains to Gram-negative strains. Your *Enterococcus faecalis* isolate is sensitive to vancomycin but the *Enterococcus faecium* vancomycin strain should be resistant – however upon testing it is also sensitive to vancomycin (needs to be addressed). Would add in oxacillin results to show that the genotype of the MRSA and MSSA is correct. Would be good to have the ug/ml of each MIC shown along with uM?

Answer 9: Thank you very much for the careful review. We sincerely apologize for the writing error, and we have rectified the inaccuracies. The VRE strain we utilized is a clinical validated vancomycin resistant strain, and we have updated the MIC and uploaded the genome sequence of this strain (please see supplementary method section 7.1 and Table S11). Additionally, we have included the experiment involving oxacillin in the revised SI. We used $\mu\text{g/mL}$ for the MIC in the revised manuscript and SI.

Question 10: Polymyxin B spelt incorrectly through-out.

Answer 10: Thank you for pointing out this problem. We have corrected the spelling throughout the revised SI (please see supplementary method section 7.2-7.3 and Table S11).

Question 11: In general the methods to not describe in enough detail to repeat the experiments. Some examples are shown below.

7.2 Antimicrobial agents and medium

What are the 5 antibacterial agents? Only 3 are mentioned.

Answer 11: We sincerely appreciate your careful proofreading of the manuscript. The correct statement should read, “Antibacterial agents including daptomycin, vancomycin, polymyxin B and oxacillin”. We have made the necessary corrections in the revised SI (please see supplementary method section 7.2).

Question 12: 7.3/7.4 Two different methods for the determination of the MIC. OD and MTT assay. Which was used?

Answer 12: Thank you for your question. We've addressed the repetition in the SI by merging the contents of sections 7.3 and 7.4. Additionally, we used the MTT method for determination the MIC (please see supplementary method section 7.3).

Question 13: 7.5 – More details on the age/sex etc of the mice.

What was the vehicle?

What was the volume of the compound, vehicle or vancomycin applied to the micrfr?

How were the compound, vehicle or vancomycin applied?

3 mice per time point? Describe.

Why were MH agar plates used, and not TSA as previously described.

The way the serial dilutions are described would not dilute the cells enough to count the high numbers in day 1, 3 and 5 and 7. What was the limit of detection?

Answer 13: Thank you for pointing out these issues. We have supplemented information about the mice and the compound in the revised manuscript (6–8-week-old female Balb/c mice weighing 20 ± 2 g were used, with DMSO as a vehicle. 20 uL of the compound, vehicle, or vancomycin were dripped onto the wound, with 12 mice per group and 3 mice per time point; please see supplementary method section 7.4). We consistently used TSA plates for CFU counts, which was modified accordingly in the revised SI (please see supplementary lines 2475 to 2495).

Regarding the comment about dilution, it may have resulted from an unclear expression in the manuscript. The solution was diluted 20-fold in each step, undergoing multiple dilutions until the CFU could be counted, rather than a single 20-fold dilution. We have provided further clarification in the revised SI (please see supplementary lines 2486 to 2488).

Question 14: 8.2 raw reads should be deposited rather than assemblies, to allow independent validation of the results.

What is the source of the ATCC29213 reference? Is it using the closed published genome. Or contigs from the illumina assembly.

More detail is need in the description of the method for the generation of spontaneous resistant mutants in the MSSA background. Why was ATCC29213 chosen for this when the emphasis has been on MRSA in the introduction? Four mutants were sequenced, need to highlight the mutations present in all these isolates. In the paper, why was that mutant chosen, do not mention the other 3. What mutations are present in each sequence isolate?

Answer 14: Thank you for your suggestion. We deposited the raw reads of clinical and SA_{22-SR} strains in GenBank. The BioProject numbers were provided in the revised SI (please see supplementary method section 8.2).

In the process of SA_{22-SR} isolation, we used SA₂₉₂₁₃ as the parental strain. Consequently, we sequenced both SA₂₉₂₁₃ and SA_{22-SR} and used the contigs from the illumina assembly as the reference. The method for generating spontaneous resistant mutants has been added to the revised manuscript (please see supplementary method

section 8.1). Regarding **22**, its clinical significance primarily lies in its efficacy against MRSA. For the identification of drug targets, we usually choose strains with a clear genetic background. For example, the Newman strain (MSSA) was also used in the following paper examining the mode of action of silver against MRSA [*Nat Commun.* **12**, 3331 (2021). <https://doi.org/10.1038/s41467-021-23659-y>]. It should be emphasized that the results of this study were not affected by the use of either MRSA or SA₂₉₂₁₃. The mutations that have the potential to affect bacterial susceptibility to **22**, were not altered by the choice of reference strain. We highlighted the mutations present in all SA_{22-SR} (please see SI Table S14). We selected a strain with the fewest mutations, so as to exclude the interference of other non-shared mutations as much as possible (please see supplementary method section 8.2, Table S14).

Question 15: Table S14. Locus Tag has an asterisk but no description. Why was Newman used for the annotation?

Answer 15: Thank you for your suggestion. When selecting the reference strain for annotation, our principle is to try to annotate all mutation sites. We tried several public annotation files and found Newman to be a good fit. We have added a description of the asterisk (please see supplementary method section 8.2, Table S14).

Question 16: 8.3.1 Not enough detail.

What primers were used?

What was the method of cloning?

What is the promoter driving expression?

Answer 16: Thank you for your question. We used the anhydrotetracycline inducible pYJ335 plasmid with the xyl-tetO promoter. The information about the plasmid and the cloning method has been included in the revised SI (please see supplementary method section 8.4.1, Table S15).

Question 17: Putting an essential histidine kinase on a plasmid (what is the plasmid copy number in your hands in SA_{22-SR}?) can lead to unintended consequences through

non-native levels of expression. The method that the gene was cloned into the pYJ335 was not described. It is an Anhydrotetracycline inducible vector. Was ATc used to induce expression? Does it complement other phenotypes? Have only shown the construct in a lysostaphin assay. Sheep blood hemolysis - hla is dramatically down. Does the addition of **22** to SA29213 increase SBA hemolysis? Increased alpha toxin expression in the presence of **22**.

Answer 17: Thank you for your suggestion. pyJ335 is a high-copy plasmid, typically with about 10 to 200 copies per bacterium. We added the method of cloning into the revised SI (please see supplementary method section 8.4.1). pyJ335 is an Anhydrotetracycline inducible vector and we used Atc to induce expression.

In the revised SI, we confirmed that the overexpression of wild type *walk* restore the transcription level of genes which are positively regulated by WalkR in SA_{22-SR} (please see SI Figure S5).

We observed increased haemolysis on sheep blood agar after **22** treatment (please see SI Figure S3).

Question 18: 8.3.2 How was the RNA isolated?

How was the data normalised?

What method was used?

What were the cells grown in?

What growth stage was analysed?

How long were the cells treated with lysostaphin for?

What strains were compared?

Answer 18: Thank you for your suggestion. The detail of the method of RNA isolation, data analysis and lysostaphin treatment were provided in the revised supplementary method section 8.4.2 and 8.6.1.

Question 19: How were the strains growth (temp, shaking speed?) What concentration of compound **22**?

Why is such a long exposure to compound **22** required?

If it is activating WalK activity, would it not happen quickly?

How much aeration?

Answer 19: Thank you for your suggestion. The details of the method of lysostaphin-induced lysis assay were provided in the revised supplementary method section 8.6.1. We corrected some errors, such as the fact that we did not use aeration for the culture and did not add **22** to the initial culture. Compound **22** was added at the same time as lysostaphin (please see supplementary method section 8.6.1).

Question 20: The level of “biofilm” being formed is very low which is characteristic of some strains of *S. aureus*. What do the P values correspond to? related to the Wt+vehicle? Not explained.

Answer 20: Thank you for your suggestion. We modified the previously ambiguous results by correcting the labels of Figure 9c in the revised manuscript.

Question 21: No description of the, cloning, expression and minimal on the protein purification.

Answer 21: Thank you very much for your reminder. Indeed, there was a significant omission in the manuscript. We have included this content in the supplementary method section 8.8 of the revised SI.

Question 22: Page 20, what reports are there of WalK activators?

Answer 22: Thank you for your suggestion. This may be a misunderstanding caused by errors in English grammar in the manuscript. Our intention was to convey that there have been no reports on WalK activators. We have made the necessary correction in the revised manuscript (please see lines 464-465).

REVIEWERS' COMMENTS

Reviewer #2 (Remarks to the Author):

The authors have adjusted all the minor issues Reviewers recommended. Therefore, this work could be published in Nature Comm now.

Reviewer #3 (Remarks to the Author):

Dear Authors,

Thank you for addressing the majority of the questions that I had with the initial submission.

Line 420-423: The Scientific Reports paper cited does not show isogenic Walk mutants and also does not highlight all the mutations additional to Walk that are associated with the strains. Therefore the contribution of Walk to vancomycin reduced susceptibility cannot be stated. I would suggest critiquing papers that have recreated isogenic mutants - some potential examples are below:

<https://journals.plos.org/plospathogens/article?id=10.1371/journal.ppat.1002359>

<https://journals.asm.org/doi/10.1128/mbio.02262-23>

<https://www.frontiersin.org/journals/microbiology/articles/10.3389/fmicb.2018.02955/full>

It is very surprising that the Walk R86C mutant recreated does not have a vancomycin phenotype – which is strongly correlated with VISA, as the transcriptional profile of R86C is of a Walk down mutant. All Walk down mutants I can think of have this phenotype.

Need to include the genome sequence of the successfully constructed Walk R86C and the spontaneous revertant strains.

[Editorial note] Please note that Reviewer 2 assessed authors' responses to Reviewer 1, and they consider the concerns of Reviewer 1 sufficiently addressed.

REVIEWER COMMENTS

Reviewer #1 (Remarks to the Author):

Reviewer 2 assessed authors' responses to Reviewer 1, and they consider the concerns of Reviewer 1 sufficiently addressed.

Reviewer #2 (Remarks to the Author):

The authors have adjusted all the minor issues Reviewers recommended. Therefore, this work could be published in Nature Comm now.

Reviewer #3 (Remarks to the Author):

Thank you for addressing the majority of the questions that I had with the initial submission.

Line 420-423: The Scientific Reports paper cited does not show isogenic WalK mutants and also does not highlight all the mutations additional to WalK that are associated with the strains. Therefore the contribution of WalK to vancomycin reduced susceptibility cannot be stated. I would suggest critiquing papers that have recreated isogenic mutants - some potential examples are below:

<https://journals.plos.org/plospathogens/article?id=10.1371/journal.ppat.1002359>

<https://journals.asm.org/doi/10.1128/mbio.02262-23>

<https://www.frontiersin.org/journals/microbiology/articles/10.3389/fmicb.2018.02955/full>

It is very surprising that the WalK R86C mutant recreated does not have a vancomycin phenotype – which is strongly correlated with VISA, as the transcriptional profile of R86C is of a WalK down mutant. All WalK down mutants I can think of have this phenotype.

Need to include the genome sequence of the successfully constructed WalK R86C and the spontaneous revertant strains.

Response Letter for Manuscript “Asymmetric Total Synthesis of Polycyclic Xanthenes and Discovery of the First WalK Activator with Potent Activity against MRSA (NCOMMS-23-36758A)”

We highly appreciate the referees for their constructive and detailed reviews. We have revised the manuscript in accordance with all their comments. The positive changes in the manuscript have been marked with track changes. We have so indicated in our point-by-point response and revision summary below.

I Response to Reviewer 1:

Comments: Reviewer 2 assessed authors' responses to Reviewer 1, and they consider the concerns of Reviewer 1 sufficiently addressed.

Answer: We appreciate reviewer 2's thorough evaluation of our response to Reviewer 1's comments and suggestions.

II Response to Reviewer 2:

Comments: The authors have adjusted all the minor issues Reviewers recommended. Therefore, this work could be published in Nature Comm now.

Answer: We appreciate the reviewer's recommendation for the publication of this work.

III Response to Reviewer 3:

Comments: Thank you for addressing the majority of the questions that I had with the initial submission.

Line 420-423: The Scientific Reports paper cited does not show isogenic WalK mutants and also does not highlight all the mutations additional to WalK that are associated with the strains. Therefore the contribution of WalK to vancomycin

reduced susceptibility cannot be stated. I would suggest critiquing papers that have recreated isogenic mutants - some potential examples are below:

<https://journals.plos.org/plospathogens/article?id=10.1371/journal.ppat.1002359>

<https://journals.asm.org/doi/10.1128/mbio.02262-23>

<https://www.frontiersin.org/journals/microbiology/articles/10.3389/fmicb.2018.02955/full>

It is very surprising that the Walk R86C mutant recreated does not have a vancomycin phenotype – which is strongly correlated with VISA, as the transcriptional profile of R86C is of a Walk down mutant. All Walk down mutants I can think of have this phenotype.

Need to include the genome sequence of the successfully constructed Walk R86C and the spontaneous revertant strains.

Answer: We highly appreciate the reviewer's comments and suggestions on our manuscript.

Question 1: Line 420-423: The Scientific Reports paper cited does not show isogenic Walk mutants and also does not highlight all the mutations additional to Walk that are associated with the strains. Therefore the contribution of Walk to vancomycin reduced susceptibility cannot be stated. I would suggest critiquing papers that have recreated isogenic mutants - some potential examples are below:

<https://journals.plos.org/plospathogens/article?id=10.1371/journal.ppat.1002359>

<https://journals.asm.org/doi/10.1128/mbio.02262-23>

<https://www.frontiersin.org/journals/microbiology/articles/10.3389/fmicb.2018.02955/full>

Answer 1: Thank you for pointing out this problem. We fully agree with your viewpoint. The three studies you mentioned that have recreated isogenic mutants are very important. We have cited these papers and provided a re-description in the revised manuscript. Please see lines 376 to 378.

Question 2: It is very surprising that the Walk R86C mutant recreated does not have

a vancomycin phenotype – which is strongly correlated with VISA, as the transcriptional profile of R86C is of a WalK down mutant. All WalK down mutants I can think of have this phenotype.

Answer 2: We quite agree with you regarding the relationship between WalK mutations and vancomycin phenotypes. Previous studies have shown that mutations at different sites in WalK have variable effects on vancomycin susceptibility (MIC ranges from 2 to 32 µg/mL, as we showed in the table below). From the table below, it can be seen that when WalK carries both R86C and I287T mutations, the MIC to vancomycin is 32 µg/mL. Our study found that when only R86C mutation is present, the MIC changes from 1 to 2 µg/mL, indicating that I287T mutation could play a stronger role in vancomycin resistance. In addition, WalK L7Q and Y225N mutations can also only cause MIC change from 1 to 2 µg/mL, such as R86C mutation alone. In addition, we did not see a large change of vancomycin MIC in SA₂₂-SR-1 strain carrying the R86C mutation. Therefore, we confirmed that the R86C single mutation did not exert a large effect on vancomycin MIC (only 2 times fold change).

Figure redacted

Ishii K, Tabuchi F, Matsuo M, Tatsuno K, Sato T, Okazaki M, Hamamoto H, Matsumoto Y, Kaito C, Aoyagi T, Hiramatsu K, Kaku M, Moriya K, Sekimizu K. Phenotypic and genomic comparisons of highly vancomycin-resistant *Staphylococcus aureus* strains developed from multiple clinical MRSA strains by in vitro mutagenesis. *Sci Rep.* 2015;5:17092.

Figure redacted

Yin Y, Chen H, Li S, Gao H, Sun S, Li H, Wang R, Jin L, Liu Y, Wang H. Daptomycin resistance in methicillin-resistant *Staphylococcus aureus* is conferred by IS256 insertion in the promoter of *mprF* along with mutations in *mprF* and *walK*. *Int J Antimicrob Agents*. 2019;54(6):673-680.

Question 3: Need to include the genome sequence of the successfully constructed WalK R86C and the spontaneous revertant strains.

Answer 3: We added the genome sequences of the successfully constructed WalK_{R86C} mutant and the spontaneous revertant strains to the revised manuscript, please see Methods ‘CRISPR-Cas9 gene editing’ section (BioProject number of constructed WalK_{R86C} mutant and spontaneous revertant strains: PRJNA1113534 and PRJNA1113431).